# Divergent molecular signatures in fish Bouncer proteins define cross-fertilization boundaries

Krista R. B. Gert [1,2], Karin Panser[1], Joachim Surm [3], Benjamin S. Steinmetz[1,7], Alexander Schleiffer [1], Luca Jovine [4], Yehu Moran [3], Fyodor Kondrashov[5,6] & Andrea Pauli [1]✉

Molecular compatibility between gametes is a prerequisite for successful fertilization. As long as a sperm and egg can recognize and bind each other via their surface proteins, gamete fusion may occur even between members of separate species, resulting in hybrids that can impact speciation. The egg membrane protein Bouncer confers species specificity to gamete interactions between medaka and zebrafish, preventing their cross-fertilization. Here, we leverage this specificity to uncover distinct amino acid residues and N-glycosylation patterns that differentially influence the function of medaka and zebrafish Bouncer and contribute to cross-species incompatibility. Curiously, in contrast to the specificity observed for medaka and zebrafish Bouncer, seahorse and fugu Bouncer are compatible with both zebrafish and medaka sperm, in line with the pervasive purifying selection that dominates Bouncer's evolution. The Bouncer-sperm interaction is therefore the product of seemingly opposing evolutionary forces that, for some species, restrict fertilization to closely related fish, and for others, allow broad gamete compatibility that enables hybridization.

Fertilization is an unequivocally key process for sexual reproduction, but our current understanding of gamete interaction and fusion is limited. Though studies over the past 20 years have identified several proteins essential for sperm-egg interaction, we lack a basic understanding of the molecular mechanisms and interaction partners for most factors, particularly in vertebrates. The only gamete fusion protein that has been discovered to date is HAP2/GCS1, an ancient eukaryotic fusogen required for fertilization in plants, algae, and apicomplexans but absent in vertebrates[1–5]. The obscurity of the vertebrate gamete fusion mechanism is compounded by the fact that only one mammalian sperm-egg interaction protein pair has been

identified: IZUMO1 on sperm interacts with egg membrane-anchored JUNO to enable binding[6,7]. Though additional essential factors including Dcst1/2, Spaca6, TMEM95, FIMP, SOF1, and Bouncer have recently been discovered[8–16], their precise roles and interaction partners have yet to be described. Functional studies pinpointing important protein domains and molecular features of individual fertility factors are therefore crucial for understanding the mechanism of fertilization and can aid in the search for their interaction partners.

One important feature of many known gamete recognition proteins is species specificity (reviewed in ref. 17). Compatibility between gametes is critical for successful sperm-egg binding and fusion, but

[1]Research Institute of Molecular Pathology (IMP), Vienna BioCenter (VBC), Campus-Vienna-Biocenter 1, 1030 Vienna, Austria. [2]Vienna BioCenter PhD Program, Doctoral School of the University of Vienna and Medical University of Vienna, 1030 Vienna, Austria. [3]Department of Ecology, Evolution and Behavior, Alexander Silberman Institute of Life Sciences, The Hebrew University of Jerusalem, 9190401 Jerusalem, Israel. [4]Department of Biosciences and Nutrition, Karolinska Institutet, Huddinge, Sweden. [5]Institute of Science and Technology Austria, Klosterneuburg, Austria. [6]Evolutionary and Synthetic Biology Unit, Okinawa Institute of Science and Technology Graduate University, Okinawa, Japan. [7]Present address: Institute of Molecular Systems Biology, Department of Biology, ETH Zürich, 8093 Zürich, Switzerland. ✉e-mail: andrea.pauli@imp.ac.at

specificity is equally important for keeping fertilization restricted to members of a single species. In animals, two classic examples of species-specific sperm-egg interactors are Bindin and EBR1 in sea urchin[18–20] and lysin and VERL in abalone[21–24]. As broadcast spawners, these marine invertebrates rely on species specificity at the level of gamete interaction to avoid hybridization with other species that might be maladaptive. Because their eggs and sperm are at risk of encountering gametes from other abalone or sea urchin species within the same geographic range, a molecular block to cross-fertilization is therefore critical in the absence of other forms of pre-zygotic reproductive isolation.

In contrast, vertebrates such as fish and mammals have both anatomical and behavioral premating reproductive barriers that come into play prior to sperm-egg interaction. In addition to premating reproductive isolation, mammals have species-specific protein interactions between sperm and the zona pellucida (ZP), a glycoprotein matrix that surrounds the egg and is considered to act as a barrier to cross-species fertilization[25,26]. Studies exploiting the taxon specificity of human sperm binding to the ZP demonstrated that 32–34 amino acids at the N-terminus of one of the constituent ZP proteins, ZP2, is both necessary and sufficient for human sperm to bind to an otherwise mouse-derived ZP[27,28]. Though fish eggs do have a protective envelope surrounding the egg, the chorion, it contains a small opening, the micropyle, that allows direct contact of sperm with the egg membrane[17,29]. We previously showed that the egg membrane protein Bouncer (Bncr) is enriched at the micropyle and is not only required for sperm binding and entry in zebrafish eggs, but also is species-specific for medaka and zebrafish, two species that diverged ~160 MYA, do not interbreed, and cannot cross-fertilize in vitro[15,30,31]. Expression of medaka Bncr in zebrafish *bncr*[−/−] eggs enables fertilization by medaka sperm but not by zebrafish sperm[15]. Similarly, expression of zebrafish Bncr in medaka eggs is sufficient for zebrafish sperm binding and fusion when these eggs are activated artificially after sperm addition[32]. Importantly, Bncr provides specificity to the interaction of the egg and sperm membranes themselves, while previously described sperm-egg interactors in marine invertebrates and mammals mediate specificity at the level of sperm interaction with the egg coat or ZP. Thus, in the absence of an outer layer conferring selectivity, Bncr may act analogously in allowing binding of only conspecific sperm to the egg membrane.

In this study, we investigate whether other fish species' Bncr proteins also mediate species-specific gamete interaction and seek to identify the molecular determinants in Bncr that mediate its specificity between zebrafish and medaka sperm. Our findings reveal important insights into the interplay between Bncr-mediated gamete compatibility and other mechanisms of reproductive isolation in fish, providing a possible explanation for the high frequency of fish hybrids in nature and the ability of certain distantly related fish species to hybridize[33,34].

## Results

### Medaka Bncra, but not Bncrb, is required on the medaka egg for fertilization

Bncr was originally identified and characterized in zebrafish[15], in which it exists as a single-exon gene (Fig. 1A, left). In medaka, however, it was unknown whether Bncr is also required for fertilization. Unlike the zebrafish locus, the medaka *bncr* locus (Fig. 1A, right) gives rise to two Bncr splice isoforms whose mature proteins are encoded by different exons, yet both adopt the characteristic Ly6/uPAR (LU) three-finger fold due to 8–10 invariant cysteines[35] (Fig. 1B). We therefore designated the two medaka proteins Bncra and Bncrb (Fig. 1A, right). The mature domain of medaka Bncra shares 38.8% identity with zebrafish Bncr and contains a predicted GPI anchor site on its C-terminus, like zebrafish Bncr, while medaka Bncrb lacks these C-terminal features (Fig. 1A, B). Though absent in zebrafish, Bncrb is conserved in many other fish species (Fig. 1B and Supplementary Fig. 1A).

To explore their potential roles in medaka fertilization, we generated exon-specific CRISPR/Cas9 mutants of *bncra* and *bncrb*, which are both highly expressed in the medaka ovary[36] (Fig. 1C). Both exon-specific mutations (Fig. 1A) resulted in frameshifts leading to premature termination codons (Supplementary Data File 1). Crosses between medaka *bncra*[−/−] females and wild-type males revealed a similar phenotype as observed in zebrafish: Bncra-deficient medaka eggs were neither activated nor fertilized, while medaka *bncra*[−/−] males were fertile when crossed to wild-type females (Fig. 1D). Moreover, the sterility of medaka *bncra*[−/−] females could be rescued by an *actin* promoter-driven, GFP-tagged medaka *bncra* cDNA transgene (Fig. 1D).

Unlike zebrafish eggs which activate upon water exposure, medaka eggs activate upon sperm binding[37,38]. Thus, the activation defect of medaka *bncra*[−/−] eggs is consistent with the sperm binding defect seen in zebrafish *bncr*[−/−] eggs[15]. In contrast to medaka *bncra*[−/−] females, medaka *bncrb*[−/−] females and males exhibited normal activation of eggs and fertility when crossed to wild-type fish (Fig. 1D). In line with its lack of predicted membrane anchorage, GFP-tagged medaka Bncrb was secreted into the perivitelline space when expressed as a cDNA transgene in zebrafish eggs, unlike membrane-localized, GFP-tagged medaka Bncra (Supplementary Fig. 1B). Although Bncrb was not essential for fertility, its function may be redundant with Bncra. To test this possibility, we performed in vitro fertilization (IVF) experiments with zebrafish *bncr*[−/−] eggs expressing a cDNA transgene encoding GFP-tagged medaka Bncrb. Neither zebrafish nor medaka sperm were able to fertilize these eggs (Supplementary Fig. 1C), demonstrating that medaka Bncrb cannot rescue fertilization in zebrafish. Bncra (hereafter, Bncr), but not Bncrb, is required for fertilization in medaka and is therefore homologous to zebrafish Bncr both in sequence and in function.

### Medaka and zebrafish sperm are compatible with multiple Bncr orthologs

Because Bncr mediates species-specific gamete interaction between medaka and zebrafish[15,32], we investigated whether other fish Bncr orthologs also show evidence for species specificity by testing their compatibility with zebrafish and medaka sperm. To test Bncr proteins over a broad evolutionary range, we generated transgenic zebrafish lines expressing Bncr homologs of common carp (*Cyprinus carpio*), tiger tail seahorse (*Hippocampus comes*), and fugu (*Takifugu rubripes*) in a *bncr* mutant background. The phylogenetic relationships among these species are depicted in Supplementary Fig. 2A. Because *bncr* transcript level correlates with fertilization rate[15], we used the *actin* promoter to drive higher egg expression of all transgenes in this study compared to the previously used *ubiquitin* promoter, including a new *actin* promoter-driven medaka Bncr line[15,39].

Concomitant with increased transgene expression (Supplementary Fig. 2B), the *actin* promoter-driven medaka Bncr line exhibited a higher average in vitro fertilization rate with medaka sperm than the *ubiquitin* promoter-driven line (55.6% vs. 5.7%[15]) (Fig. 2A). Increased medaka Bncr expression in the egg further resulted in higher average in vivo (32.2%) and in vitro (4.2%) fertilization rates with zebrafish sperm (Supplementary Fig. 2C and Fig. 2A), indicating that medaka Bncr's specificity for medaka sperm can be partially overridden by overexpression. Importantly, however, medaka sperm remain unable to fertilize zebrafish eggs overexpressing zebrafish Bncr at a similar level (Fig. 2A and Supplementary Fig. 2B). This suggests that the species specificity of Bncr is asymmetrical and could be governed by different features in Bncr for zebrafish vs. medaka sperm. Because all the tested transgenes were expressed on zebrafish eggs, there may be intrinsic bias for zebrafish sperm given conspecificity of the egg and the possible influence of other, currently uncharacterized sperm-egg interactors that would be species-matched for zebrafish but not medaka sperm. However, while medaka sperm are strictly

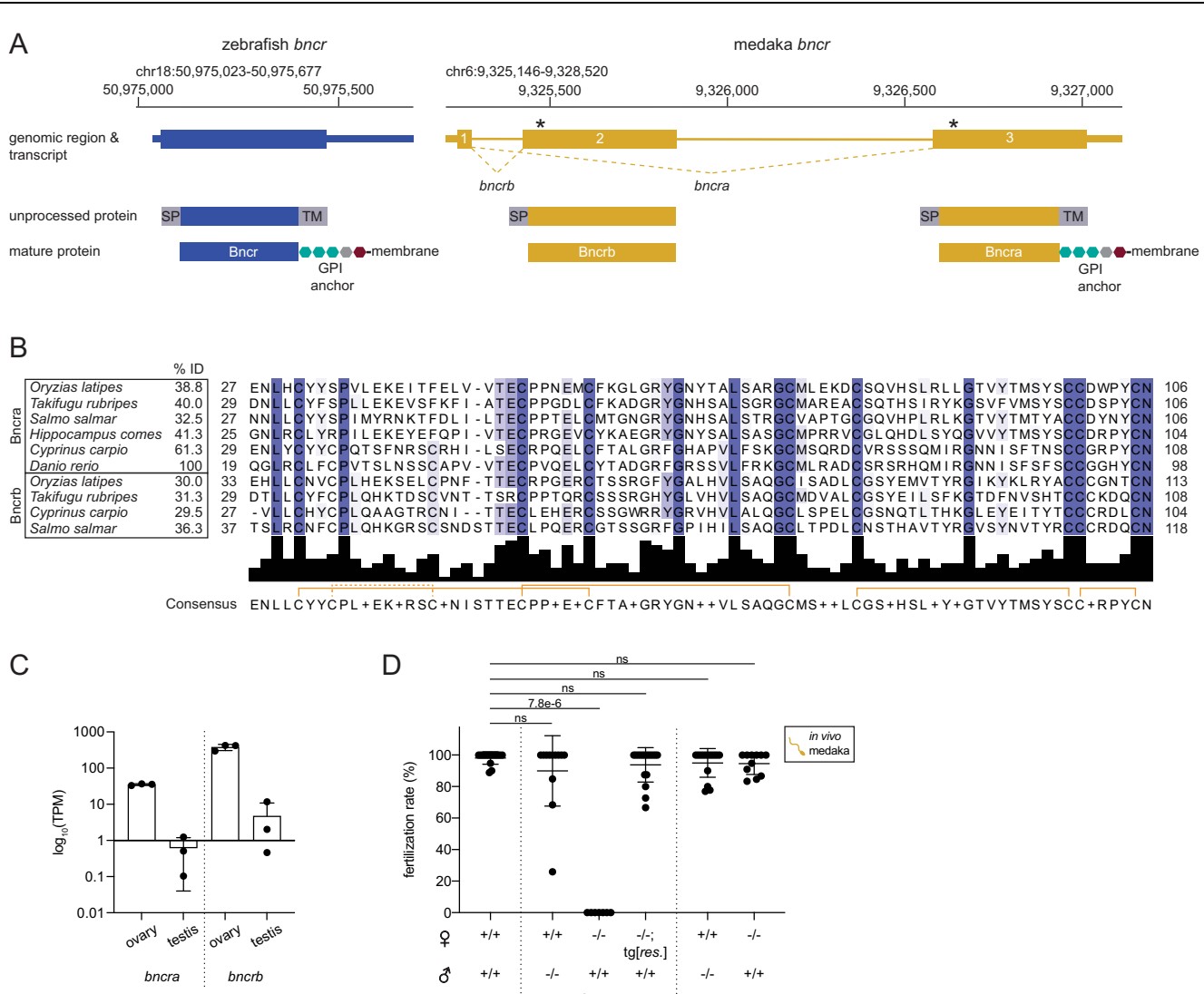

**Fig. 1 | Medaka Bncra, but not Bncrb, is required on the medaka egg for fertilization. A** Genomic regions and resulting transcripts and proteins of the *bncr* locus in zebrafish (blue; GRCz11/danRer11) and medaka (yellow; Ensembl 93: Jul 2018 (GRCh38.p12)). Zebrafish Bncr is encoded by a single-exon gene (NM_001365726.1). The medaka *bncr* locus (ENSORLG00000004579) comprises three exons that are alternatively spliced to generate Bncra (exons 1 and 3; ENSORLT00000005754) and Bncrb (exons 1 and 2; ENSORLT00000005758). The location of the CRISPR-induced genomic deletions for medaka *bncra* (5-nt deletion in exon 3) and *bncrb* (38-nt deletion in exon 2) are indicated by asterisks. The gene structures are depicted with untranslated regions (thin rectangles) and coding sequences (thick rectangles). **B** Protein sequence alignment of the LU domains of Bncra and Bncrb from selected fish species (Supplementary Data File 2). Note that seahorse Bncrb is a predicted translation product from a genomic region. Purple shading indicates amino acids with at least 30% conservation. The percent amino acid sequence identity (% ID) within the mature domains is indicated. Disulfide bonds are indicated by orange brackets. **C** Expression values of *bncra* and *bncrb* transcripts in medaka ovary and testis based on RNA-seq[36] from three biological replicates for each tissue. The Y-axis is plotted in $\log_{10}$ scale. TPM transcripts per million. **D** Quantification of in vivo fertilization rates from wild type and medaka *bncra* and *bncrb* mutants; tg[res], transgenic rescue. (Kruskal–Wallis test with Dunn's multiple comparisons test; ns not significant). Means ± SD are indicated in (**C**) and (**D**).

incompatible with zebrafish Bncr regardless of expression level, zebrafish sperm can fertilize both artificially activated wild-type (2% on average) and zebrafish Bncr-expressing (24.3% on average) medaka eggs[32], suggesting a potentially lower degree of stringency in zebrafish sperm-Bncr interaction compared to that of medaka.

Contrary to the hypothesis that Bncr is generally species-specific among fish, zebrafish and medaka sperm were compatible with multiple Bncr proteins (Fig. 2). Carp Bncr, being 61.3% identical to zebrafish Bncr, showed complete incompatibility with medaka sperm, yet rescued fertilization in vivo and in vitro with zebrafish sperm (Supplementary Fig. 2C and Fig. 2A). Unexpectedly, despite their high identity with medaka Bncr, seahorse and fugu Bncr were compatible with zebrafish sperm in vivo and

in vitro (Supplementary Fig. 2C and Fig. 2A), and surprisingly, were also compatible with medaka sperm in vitro (Fig. 2A). We assessed the relative bias of the tested Bncr proteins for zebrafish vs. medaka sperm by calculating the bias index (Fig. 2B) using the IVF data (Fig. 2A) for each line. While zebrafish and carp Bncr strictly favor zebrafish sperm, seahorse and medaka Bncr display bias for medaka sperm (Fig. 2B). Interestingly, fugu Bncr does not exhibit bias for either sperm (Fig. 2B). These results further support the idea that zebrafish sperm interact more indiscriminately with Bncr proteins than medaka sperm. Importantly, because seahorse and fugu Bncr exhibit dual compatibility, the features required for successful interaction with both medaka and zebrafish sperm can functionally coexist within the same Bncr protein.

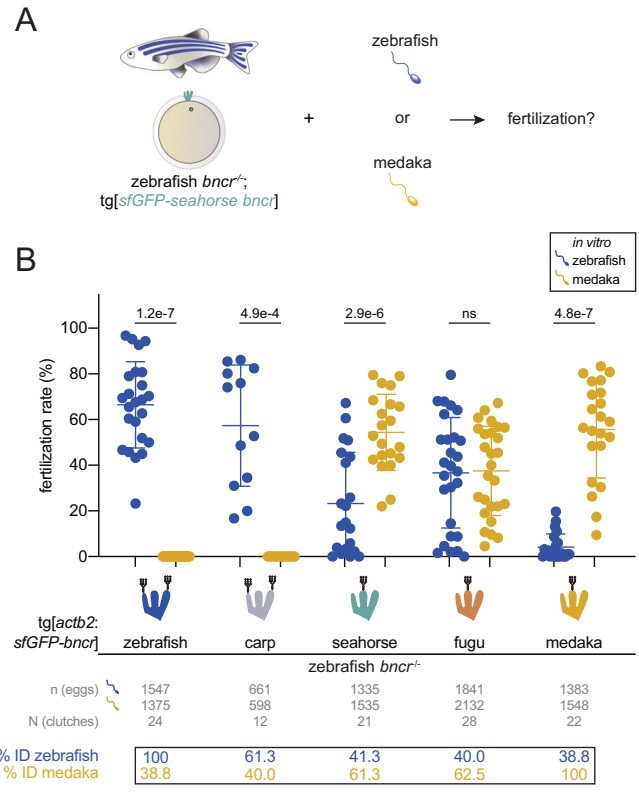

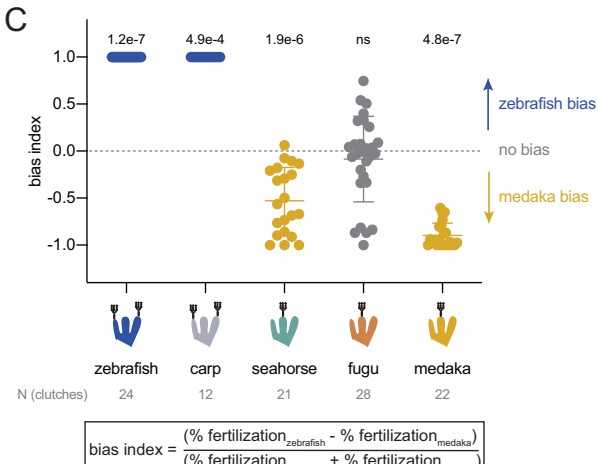

**Fig. 2 | Medaka and zebrafish sperm are compatible with multiple Bncr orthologs. A** Experimental setup for performing comparative medaka/zebrafish IVF with transgenic zebrafish *bncr*⁻/⁻ eggs expressing different fish Bncr orthologs. **B** IVF data obtained from transgenic zebrafish *bncr*⁻/⁻ lines expressing either zebrafish, carp, seahorse, fugu, or medaka Bncr with medaka vs. zebrafish sperm. N-glycans are depicted on each Bncr ortholog as sugar chain symbols on each glycosylated finger. (Two-tailed Wilcoxon matched-pairs signed rank test with the method of Pratt; ns not significant). **C** Plot of the bias index values derived from the IVF data in (**A**). The formula for the bias index is shown. (Two-tailed Wilcoxon signed rank test vs. theoretical median of 0 with the method of Pratt). Means ± SD are indicated in (**B**) and (**C**).

## Medaka/zebrafish Bncr chimeras reveal specificity determinants in fingers 2 and 3

To uncover the molecular basis for this species-specific asymmetry, we investigated which parts of the Bncr protein (referred to as "fingers" given Bncr's three-finger fold[15,35]) confer medaka/zebrafish specificity. To this end, we generated a set of transgenic zebrafish lines that

express medaka/zebrafish Bncr chimeras in the zebrafish *bncr*⁻/⁻ background. These chimeras comprise eight different combinations of fingers as well as the upper ("top," all three fingers excluding the "base") and lower ("base") regions of medaka and zebrafish Bncr (Fig. 3A and Supplementary Data File 3).

By performing IVF experiments with these chimeric Bncr lines, we systematically tested the role of each finger or combination of fingers for compatibility with medaka vs. zebrafish sperm (Fig. 3A). Changing only the "top" but not the "base" to the medaka sequence enabled fertilization by medaka sperm and abrogated fertilization by zebrafish sperm, revealing that the species specificity determinants are encoded within the upper regions of the three fingers (Fig. 3B). Single medaka finger substitutions were not sufficient to rescue fertilization with medaka sperm. Changing finger 3 to medaka greatly decreased fertilization rates with zebrafish sperm in vitro, suggesting a role for finger 3 in mediating specificity (Fig. 3B), though fertilization rates in vivo remained high (72.7% on average) (Supplementary Fig. 3A). Combinations of medaka fingers 1 + 2 and 2 + 3 were compatible with both species' sperm. Combining medaka fingers 1 + 3 failed to rescue fertilization with both sperm in vitro (Fig. 3B) despite low in vivo fertilization rates (2.5% on average) with zebrafish sperm (Supplementary Fig. 3A) and expression on the egg membrane (Supplementary Fig. 3B). Though compatible with both species' sperm, the medaka finger 1 + 2 chimera showed a clear bias for zebrafish sperm (Fig. 3C). Bias for medaka over zebrafish sperm was evident only upon changing fingers 2 + 3 together or all three (top) to the medaka sequence (Fig. 3C). These data demonstrate a requirement for medaka finger 2 in addition to either finger 1 or 3 for medaka sperm compatibility, with finger 3 having a stronger effect in shifting bias toward medaka sperm. In addition, chimeras containing zebrafish finger 3 maintain a bias for zebrafish sperm, further underscoring a role for finger 3 in determining species specificity. These results hint toward clarifying the asymmetric requirements in Bncr for medaka vs. zebrafish sperm: while medaka require features in both fingers 2 + 3 for specificity, only finger 3 is required for zebrafish specificity.

## A positively selected, Oryzias-specific change hampers zebrafish sperm compatibility

To identify more precisely the features within medaka and zebrafish Bncr that underlie the incompatibility between these two species' gametes, we used evolutionary approaches. First, using fish Bncr phylogeny, we predicted ancestral states of the Bncr protein (see "Methods") that contain the predicted changes undergone between the zebrafish and medaka orthologs (Fig. 4A, B and Supplementary Data File 3). To identify when in Bncr's evolutionary history incompatibility with zebrafish sperm may have arisen and which amino acid changes caused this, we generated transgenic lines in the zebrafish *bncr*⁻/⁻ background expressing the predicted ancestral states of Bncr between seahorse and fugu Bncr and tested them for fertility with zebrafish and medaka sperm. In line with the dual compatibility observed for seahorse and fugu Bncr, ancestral states at nodes A–D (the same sequence was predicted for these four nodes), E, and G exhibited compatibility for both species' sperm (Fig. 4B, C). Nodes A–D and G, however, rescued poorly with both zebrafish and medaka sperm despite expression at the egg membrane (Supplementary Fig. 4A, B) and the ability to rescue fertilization in vivo with zebrafish sperm (Supplementary Fig. 4A, B), suggesting that these Bncr states contain features detrimental for interaction with both sperm. While the ancestral Bncr at node E showed similar compatibility with both zebrafish and medaka sperm in vitro, a clear bias for medaka sperm was observed at node F which immediately precedes the *Oryzias* (medaka) genus clade (Fig. 4C, D), pointing toward the presence of an *Oryzias*-specific change that hinders zebrafish compatibility.

In a second evolutionary approach, we performed positive selection analysis of the mature domain of fish Bncr proteins (see

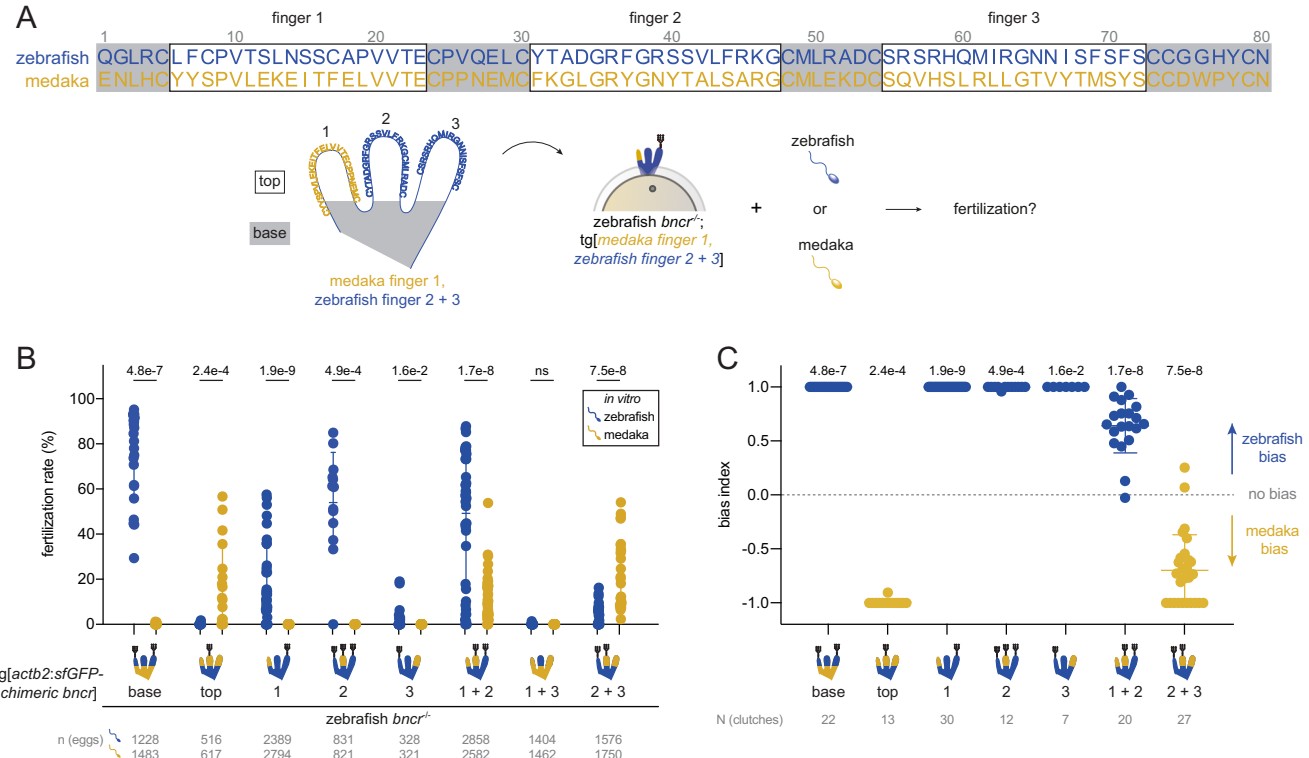

**Fig. 3 | Medaka/zebrafish Bncr chimeras reveal specificity determinants in fingers 2 and 3. A** Zebrafish (blue) and medaka (dark yellow) mature Bncr protein sequence alignment and schematic of the Bncr protein fold. Fingers are labeled 1, 2, and 3 and correspond to the amino acids in boxes in the protein sequence alignment. Amino acid sites are numbered starting with the first site in the mature protein. Note that each finger is bounded by cysteine residues that keep disulfide bridges intact. "Top" and "base" are indicated. **B** Comparative medaka/zebrafish

IVF data with Bncr chimera lines. N-glycosylation pattern of each chimera is depicted with sugar chain symbols. Means ± SD are indicated. (Two-tailed Wilcoxon matched-pairs signed rank test with the method of Pratt; ns not significant). **C** Plot of the bias index values derived from the IVF data in (**B**). Bias could not be calculated for data pairs for which the fertilization rates with both sperm were equal to 0. Means ± SD are indicated. (Two-tailed Wilcoxon signed rank test vs. theoretical median of 0 with the method of Pratt).

"Methods"). Positively selected (variable) amino acids indicate evolutionary pressure to diversify a protein sequence, while negatively selected (conserved) amino acids are a signature of resistance to change. Positive selection analyses revealed that the majority of Bncr's codons (52 out of 87) are evolving under pervasive purifying (negative) selection in the tested fish species, indicating evolutionary pressure to conserve the amino acid sequence and thereby preserve binding interactions (Supplementary Data File 4). However, our analyses found evidence for positive selection at two sites within Bncr. Site 15 (Ser in zebrafish; Ile in medaka) in finger 1 had signatures of pervasive diversifying (positive) selection throughout the phylogeny of tested fish Bncr proteins. In contrast, site 63 (Arg in zebrafish; Leu in medaka) in finger 3 had evidence of episodic diversifying selection specifically in the *Oryzias* lineage (Fig. 4A, Supplementary Fig. 4C and Supplementary Data File 4), representing a medaka-specific change that could influence Bncr incompatibility between medaka and zebrafish. Both positively selected sites differed between the ancestral Bncr sequences at nodes E and F, concomitant with a switch in bias from zebrafish to medaka sperm (Fig. 4A, D) and suggesting a possible contribution to the observed species specificity.

Based on our evolutionary analyses, we tested the contribution of sites 15 and 63 in determining medaka/zebrafish specificity. Moreover, given the chimera data that implicated finger 2 in medaka sperm compatibility, we further compared medaka-compatible vs. incompatible Bncr sequences and identified site 45 as another candidate that might contribute to specificity (Arg in zebrafish; Ala or Gly in all medaka-compatible sequences) (Fig. 4A and Supplementary Data File 3). Based on the AlphaFold structural predictions[40,41] of zebrafish and medaka Bncr, the two arginines in sites 45 and 63 may together

form a positively charged patch in zebrafish Bncr that is absent in medaka Bncr (Fig. 5A, B). We hypothesized that this positively charged patch may either be unfavorable for medaka sperm or beneficial for zebrafish sperm interaction.

Using transgenic lines in the zebrafish *bncr*[−/−] background, we tested whether the amino acids in these sites alone or in combination were sufficient to switch the specificity of one species' Bncr to favor the other species' sperm. Introduction of zebrafish amino acids into medaka Bncr increased compatibility with zebrafish sperm in vivo and in vitro, particularly when introduced in combination (Fig. 5C and Supplementary Fig. 5A). Substituting both A45 and L63 for R in medaka Bncr was sufficient to cause a clear shift in bias toward zebrafish sperm, suggesting that the positively charged patch mediated by these arginine residues is beneficial for zebrafish sperm interaction (Fig. 5D). In contrast, none of the tested medaka amino acid substitutions in zebrafish Bncr were sufficient to enable fertilization by medaka sperm, and neither did they disrupt compatibility with zebrafish sperm (Fig. 5E and Supplementary Fig. 5B, C). Because medaka sperm retained compatibility with all medaka Bncr substitution mutants, other features within Bncr are required to determine medaka specificity (Fig. 5C).

## Medaka Bouncer requires N-glycosylation in finger 2

To uncover these features, we examined all tested sequences (fish Bncr orthologs, ancestral states, and medaka/zebrafish Bncr chimeras) for elements that are always present when compatible with one species' sperm but not both. All constructs that can rescue fertilization with medaka sperm contain a single predicted N-glycosylation site in finger 2 (NXS/T, where X is any amino acid except proline), while zebrafish and carp Bncr contain N-glycosylation sites in fingers 1 and 3 (Fig. 4A).

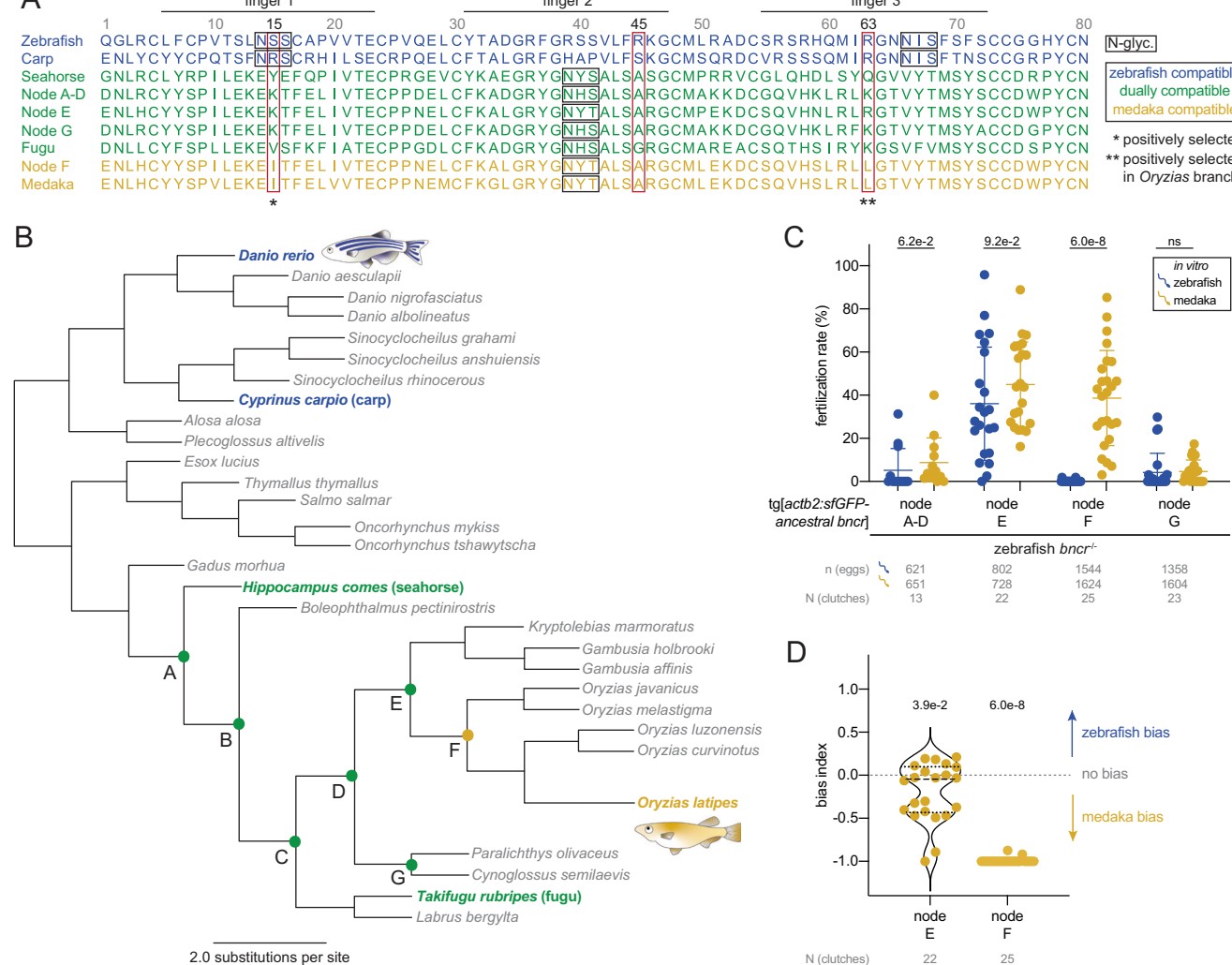

**Fig. 4 | A positively selected, *Oryzias*-specific amino acid change hampers zebrafish sperm compatibility. A** Protein sequence alignment of fish Bncr orthologs and predicted ancestral states of Bncr. Fingers 1, 2, and 3 are indicated. Amino acid sites are numbered starting with the first site in the mature protein. Zebrafish-compatible sequences are blue, medaka-compatible sequences are yellow, and dually compatible sequences are green. Red rectangles demarcate sites 15, 45, and 63 which were tested individually and in combination for their role in species specificity, while N-glycosylation sites are marked with a black rectangle (see Fig. 5). The two positively selected sites are highlighted with asterisks. **B** Phylogenetic tree of the predicted Bncr ancestral states according to fish phylogeny.

Tested nodes (A–G) are marked with a closed circle and colored according to compatibility as in (**A**). Nodes A–D were predicted to have the same sequence and are therefore equivalent. **C** Comparative medaka/zebrafish IVF data from the tested Bncr ancestral states. Means ± SD are indicated. (Two-tailed Wilcoxon matched-pairs signed rank test with the method of Pratt; ns not significant. **D** Plot of bias index values derived from the IVF data pairs from nodes E and F in (**C**). Bias was not calculated for nodes A–D and G for which the average fertilization rates with both sperm were <10%. Median (dashed line) and quartiles (dotted lines) are shown. (Two-tailed Wilcoxon signed rank test vs. theoretical median of 0 with the method of Pratt).

In line with our previous observation that non-glycosylated zebrafish Bncr is functional with zebrafish sperm[15], we hypothesized that the presence of N-glycosylation in finger 2 may contribute to the medaka-specific requirement for compatibility that is not shared by zebrafish and manifests as asymmetrical specificity. To test the role of both number and position of Bncr N-glycosylation sites in medaka/zebrafish specificity, we generated transgenic lines in the zebrafish *bncr*−/− background expressing medaka and zebrafish Bncr N-glycosylation site variants that exhibit the expected N-glycosylation patterns in western blots (Supplementary Fig. 5D, E and Supplementary Data File 3).

We found that any changes to the N-glycosylation pattern of zebrafish Bncr, even when mimicking the N-glycosylation pattern of medaka Bncr with only finger 2 glycosylated, maintained zebrafish sperm compatibility but were not sufficient to rescue medaka sperm compatibility (Fig. 5F, left and Supplementary Fig. 5F). In contrast, we observed a strict requirement for finger 2 N-glycosylation of medaka

Bncr: removal of this N-glycosylation site abrogated fertilization with both medaka and zebrafish sperm (Fig. 5F, right and Supplementary Fig. 5G). Fertilization with either sperm could not be restored by adding an N-glycosylation site to medaka Bncr on finger 1, 3, or both despite membrane expression of all constructs (Fig. 5F, right and Supplementary Fig. 5H). Addition of an N-glycosylation site to finger 3 of medaka Bncr did not abolish medaka sperm compatibility but decreased the fertilization rate, suggesting that this feature is unfavorable for medaka sperm interaction (Fig. 5F, right). We then tested whether combining both the medaka N-glycosylation pattern (finger 2) and amino acid changes (R45A and/or R63L) or the full sequence of medaka finger 3 in a zebrafish Bncr construct would be sufficient for medaka sperm compatibility. When expressed transgenically on zebrafish *bncr*−/− eggs, none of these constructs enabled fertilization with medaka sperm and rescued fertilization poorly with zebrafish sperm in vitro (Fig. 5G). Similarly, medaka Bncr variants with the zebrafish N-glycosylation pattern in combination with amino acid changes (A45R

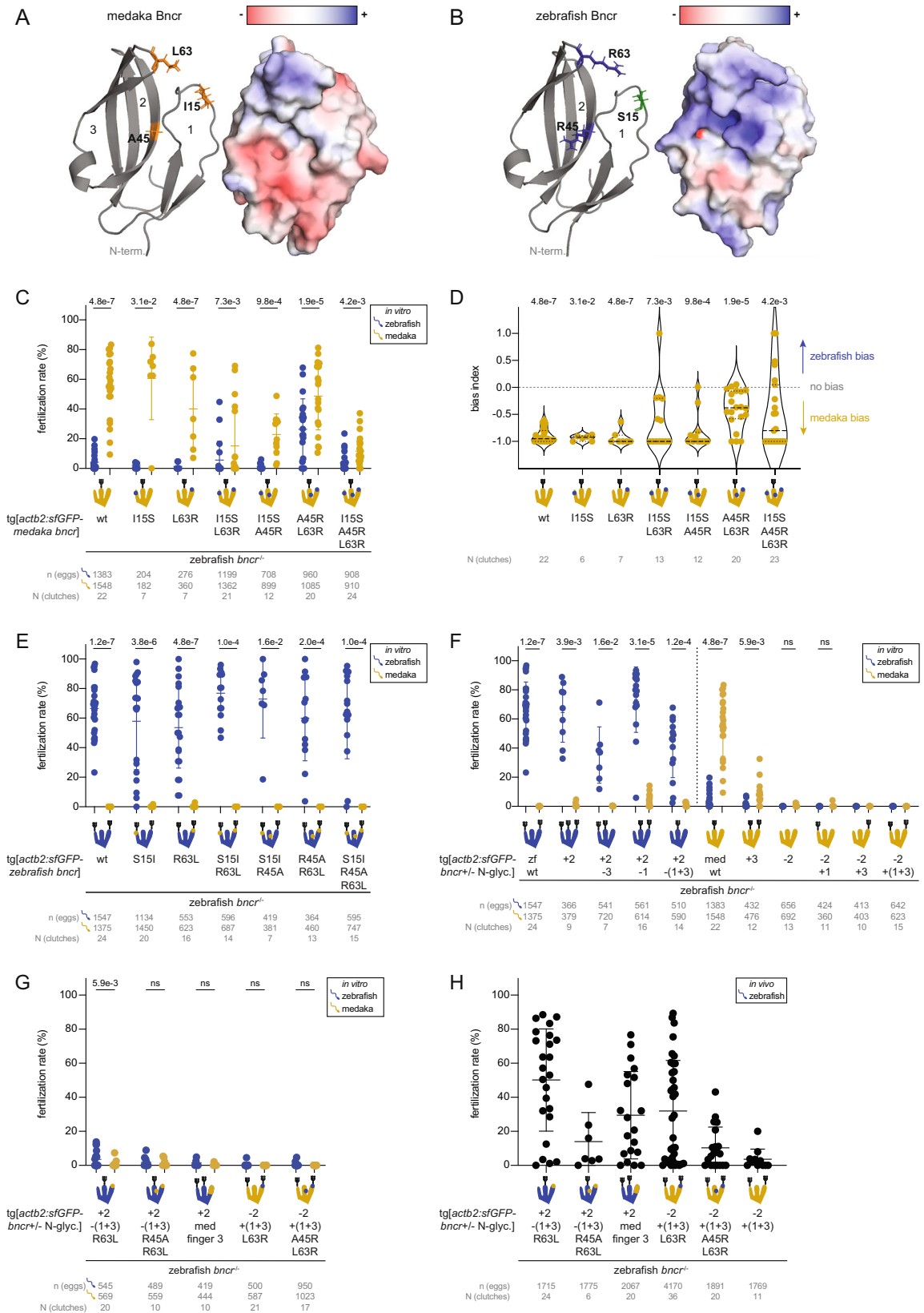

and/or L63R) failed to rescue fertilization with both medaka and zebrafish sperm in vitro despite membrane expression (Supplementary Fig. 5H), supporting the importance of finger 2 N-glycosylation for medaka Bncr function (Fig. 5G). However, while changing only the N-glycosylation pattern of medaka Bncr to that of zebrafish resulted in

poor rescue with zebrafish sperm in vivo, introducing both A45R and L63R or only L63R in addition increased fertilization rates with zebrafish sperm in vivo without visible differences in membrane expression of these constructs (Fig. 5H and Supplementary Fig. 5H). This suggests that R63 can partially reconstitute the binding site of

**Fig. 5 | Zebrafish sperm favor a positively charged Bncr surface, while medaka sperm require finger 2 N-glycosylation for compatibility.** AlphaFold-predicted models of medaka Bncr (**A**) and zebrafish Bncr (**B**) (cartoon, left; surface representation depicting electrostatics, right). Amino acids that were mutated are indicated in the model as sticks and color-coded: hydrophobic (orange), positively charged (blue), polar (green). **C** Medaka/zebrafish IVF with medaka Bncr constructs, in which individual amino acids or combinations thereof were substituted for the corresponding amino acid(s) in zebrafish Bncr. **D** Plot of bias index values derived from the IVF data in (**C**). Bias could not be calculated for data pairs for which the fertilization rate with both sperm was equal to 0. Medians (dashed lines) and quartiles (dotted lines) are shown. **E** Medaka/zebrafish IVF with zebrafish Bncr constructs, in which individual amino acids or combinations thereof were

substituted for the corresponding amino acid(s) in medaka Bncr. **F** Medaka/zebrafish IVF experiments to assess the importance of N-glycosylation in Bncr's species specificity. IVF with zebrafish Bncr N-glycosylation site variants (left); IVF with medaka Bncr N-glycosylation site variants (right). **G** Medaka/zebrafish IVF experiments testing sufficiency of N-glycosylation pattern combined with specific amino acid changes for determining Bncr's species specificity. IVF with zebrafish Bncr variants (left); IVF with medaka Bncr variants (right). **H** In vivo zebrafish fertilization rates of combined N-glycosylation and amino acid substitution Bncr variants. Means ± SD are indicated in (**C**, **E–H**). **C**, **E–G**: two-tailed Wilcoxon matched-pairs signed rank test with the method of Pratt; *p* values could not be calculated for samples in which all data points were 0. **D**: two-tailed Wilcoxon signed rank test vs. theoretical median of 0 with the method of Pratt.

zebrafish sperm on medaka Bncr, further implicating a central role for this amino acid in zebrafish sperm interaction with Bncr. Overall, these data support N-glycosylation of finger 2 in medaka Bncr as a medaka-specific requirement for sperm compatibility that is absent in medaka-incompatible Bncr proteins. However, while necessary, this N-linked glycan is not sufficient to enable medaka sperm compatibility with zebrafish Bncr even when combined with A45 and L63 or the entire medaka finger 3 sequence, underscoring the stringent specificity in place for medaka sperm-Bncr interaction compared to that of zebrafish.

## Discussion

Fertilization lies at the heart of sexual reproduction and is therefore an essential process in nearly all major groups of organisms. Our study explores the role of a Ly6/uPAR protein, Bncr, in species-specific fertilization. Punctuated by typically ten highly conserved cysteine residues, the LU domain characterizing these proteins adopts a three-finger fold stabilized by disulfide bonds[35]. The constituent fingers can vary in length and sequence composition, thereby comprising an adaptable protein module that displays great diversity in terms of tissue expression and function, ranging from immune cells to male reproductive tissues and epithelial cells with cell type-specific roles[35]. Our work here demonstrates how both specific amino acid changes and differences in N-glycosylation pattern of Bncr's three-finger domain can impose species-restricted interactions for certain species combinations, while other Bncr orthologs, despite overall sequence divergence, maintain cross-compatibility.

While orthologous fertilization proteins contain broadly conserved domains and structural folds, they are also often marked by functional or structural diversification. Bncr is no exception. In addition to the medaka-specific adaptations revealed in this study, Bncr has diverged greatly in terms of cell type expression and function between fish and mammals. SPACA4, Bncr's mammalian homolog, is expressed on sperm rather than eggs and is required for ZP binding and penetration[13], demonstrating how this flexible protein domain has been adapted for the vastly different reproductive modes of fish vs. mammals, yet retains functional importance in sperm-egg interaction in both cases. Similarly, HAP2 is homologous with viral class II fusion proteins and maintains this ancestral role in mediating gamete membrane fusion in multiple taxa[2,42–44], yet its divergent membrane insertion loops have undergone structural changes between flowering plants and trypanosomes[45]. IZUMO1, SPACA6, and TMEM95 each contain an immunoglobulin-like domain in addition to a 4-helix bundle —a structural fold that places them into a conserved superfamily of fertilization-associated proteins in vertebrates[46,47]. Spe-45 in *C. elegans* shares structural and functional homology with mouse IZUMO1[48,49], while the plant sperm membrane protein GEX2 and *Chlamydomonas* gamete attachment protein FUS1 as well as HAP2 contain domains which also adopt Ig-like folds, underscoring the broad conservation of this structural feature in fertilization proteins[50–52]. These conserved protein domains have been repeatedly adapted to the reproductive contexts of distantly related species, revealing their versatility and

highlighting common molecular themes within fertilization across phyla.

In this study, we investigated the role of Bncr in medaka and its features that determine species specificity between medaka and zebrafish sperm. Examination of the medaka *bncr* locus revealed the presence of two splice isoforms, Bncra and Bncrb, that are present in many fish species (Supplementary Fig. 1A). These splice isoforms likely arose by gene duplication, which has been shown to influence the evolution of fertilization proteins in many species (reviewed in ref. [53]) and has been implicated in Ly6/uPAR gene evolution[35]. In the case of zebrafish, however, Bncrb appears to have been lost. Characterization of Bncra and Bncrb in medaka revealed that while Bncra is required for fertilization like zebrafish Bncr, neither male nor female medaka Bncrb mutants had any apparent fertilization defects and transgenic medaka Bncrb failed to rescue fertilization with medaka sperm when expressed in zebrafish eggs (Fig. 1D and Supplementary Fig. 1C). Bncra (Bncr) is therefore conserved as an essential fertilization factor in distantly related fish species, but the precise role of Bncrb in the egg requires further study. Bncrb may support other fertilization proteins, for example in sperm chemoattraction to the egg, but it is not necessary for this process.

Our investigation into Bncr's role in mediating species-specific fertilization in fish revealed that instead of exhibiting strong selectivity for conspecific sperm as observed for zebrafish and medaka previously[15] and in this study, Bncr orthologs in general maintain more widespread compatibility among species than previously expected (Fig. 2). Indeed, the strong specificity between medaka and zebrafish Bncr-sperm pairs appears restricted to these two species but may extend to gamete interactions between fish from genus *Oryzias* and suborder *Cyprinoidei* in general given the incompatibility between medaka sperm and carp Bncr. A limitation of our study is that only medaka and zebrafish sperm were tested for compatibility with other fish Bncr proteins, necessitating future work testing a wider range of species' sperm to form a comprehensive picture of Bncr-sperm compatibility across fishes. Additionally, while this study and our previous work[15] demonstrate that Bncr is necessary for sperm binding and subsequent fertilization, other, currently unidentified factors on the sperm and egg membranes also contribute to gamete interaction and fusion. Suboptimal cross-compatibility of these factors between medaka and zebrafish may decrease the overall efficiency of medaka sperm fertilizing zebrafish eggs, however, our work clearly demonstrates that providing the zebrafish egg with a medaka-compatible Bncr can even allow medaka sperm to outperform zebrafish sperm. We have furthermore shown that zebrafish sperm gain the ability to fertilize medaka eggs upon expression of zebrafish Bncr[32]. Together, these observations indicate that Bncr is the primary source of molecular incompatibility between medaka and zebrafish gametes and that the remaining fertilization machinery is functionally conserved between them.

Two important themes emerge from our observations. First, species-specific Bncr-sperm interaction can be partially overcome by expression level, reminiscent of the concentration-dependent

interactions previously seen with lysin and Bindin[22,54]. Secondly, zebrafish sperm interact more indiscriminately with the tested Bncr homologs compared to medaka sperm, exhibiting a wider range of compatibility. This promiscuity may contribute to the observed high frequency of hybridization among species within *Cyprinidae*[55] and may in part explain the ability of *Danio* species to hybridize with one another and other species within *Cyprinidae*[56,57]. This is further underscored by the fact that unlike previously described species-specific fertilization factor pairs like Bindin/EBR1[58,59] and lysin/VERL[60,61], Bncr's evolution is marked mostly by negative rather than positive selection. By maintaining cross-species compatibility, Bncr may have played a part in allowing cross-fertilization and hybridization of diverse fish species, particularly for those without other modes of reproductive isolation.

A high degree of similarity between Bncr proteins may also allow hybridization of more distantly related species. In general, the probability of obtaining viable hybrid offspring from a given interspecies cross decreases with increasing phylogenetic distance between the parental species[62,63]. Despite this general rule, distantly related species have been observed to cross-fertilize and produce viable hybrid offspring. For example, Russian sturgeon and American paddlefish are separated by a similar phylogenetic distance as medaka and zebrafish (~180 MYR), yet in vitro cross-fertilization is possible between them and yields viable hybrids[34]. We speculate that these two species can cross-fertilize due to the high degree of identity (~92%) between their predicted mature Bncr proteins (Supplementary Data File 5). Thus, our observation that Bncr homologs from a wide phylogenetic range (Fig. 4B and Supplementary Fig. 2A) are compatible with zebrafish sperm may explain how gametic compatibility can be preserved even over large phylogenetic distances, owing to the predominant influence of negative selection on Bncr's evolution.

However, while cross-species gamete compatibility via Bncr is permissive for hybridization, other reproductive isolation mechanisms are at play within fish species, preventing widespread hybridization from occurring. Such mechanisms involve spatial, temporal, or behavioral isolation that ultimately prevent successful mating events between members of different species and thereby maintain species distinctness[64]. Speciation in Lake Victoria cichlids has been influenced by sensory drive that impacts mate choice[65,66], as opposed to prezygotic isolation via Bncr given the high sequence identity among their Bncr orthologs (Supplementary Data File 5). For instance, the co-occurring haplochromine cichlid species *Pundamilia nyererei* and *P. pundamilia* are largely reproductively isolated due to female mate preference for a specific male coloration[67]. However, increased water turbidity reduces the effect of mate choice, leading to increased hybridization between *P. nyererei* and *P. pundamilia*[68]. In this case, hybridization capacity is preserved through compatible gamete molecules including Bncr, but a sensory, behavioral mechanism has evolved to form a reproductive barrier. If such premating barriers fail due to changing environmental conditions as in this example, hybridization may still occur as long as gamete compatibility is maintained. This suggests that cross-species Bncr compatibility may be a possible contributor to the exceptional biodiversity of fish, at least in some lineages[69]. However, postzygotic incompatibilities such as hybrid inviability or sterility also influence the success of hybridization and potential for speciation, particularly for more distantly related species[63].

Our study reveals that the features in Bncr that dictate medaka or zebrafish compatibility are not mutually exclusive and comprise a different set of requirements involving a combination of specific amino acids and N-glycosylation pattern for interaction with each species' sperm. As shown for fugu and seahorse Bncr proteins, a Bncr protein can fulfill the requirements for interaction with both medaka and zebrafish sperm simultaneously. Constituent amino acids in finger 3 appear critical for maintaining successful sperm interaction for both zebrafish and medaka, but the context is decisive (Fig. 3B, C). Specifically, introducing L63 (R63 in zebrafish) into finger 3 in a medaka Bncr-like context contributes to a clear preference for medaka over zebrafish sperm as revealed by comparing fertilization rates for ancestral state nodes E and F (Fig. 4C, D). Introducing L63 into an otherwise zebrafish Bncr protein, however, is not sufficient to disrupt zebrafish compatibility nor enable medaka compatibility (Fig. 5D), indicating that additional features are required for species-specific Bncr interaction. When both A45 and L63 in medaka Bncr are mutated to R as in zebrafish Bncr, this mutant shifts in bias toward zebrafish sperm, suggesting that the positively charged surface provided by these residues is beneficial for zebrafish sperm, yet medaka sperm are not deterred from interaction (Fig. 5E).

Our analysis further revealed that N-glycosylation of Bncr is another context-dependent feature with differential influence on medaka and zebrafish sperm interaction. The unidentified Bncr interaction partner on zebrafish sperm tolerates both a lack of N-glycosylation in fingers 1 and 3[15] and the presence of N-glycosylation in finger 2 in zebrafish Bncr (Fig. 5F), lending further support to the idea that zebrafish sperm have fewer requirements for successful binding. In contrast, removal of N-glycosylation from finger 2 of medaka Bncr prevents fertilization with either sperm, and functionality cannot be rescued by addition of N-glycosylation to finger 1, 3, or both (Fig. 5F). This may be a result of failed protein folding or trafficking to the membrane, yet all medaka Bncr N-glycosylation variants were still detected at the egg membrane in transgenic zebrafish lines (Supplementary Fig. 5H). Interestingly, we observed that introduction of both A45R and L63R or L63R alone into medaka Bncr lacking finger 2 N-glycosylation was able to restore functionality of this construct with zebrafish sperm in vivo (Fig. 5H), further highlighting the importance of these amino acid sites for zebrafish sperm interaction and the higher degree of flexibility exhibited by the zebrafish interaction partner. Importantly, all medaka-compatible Bncr sequences have finger 2 N-glycosylation (Figs. 3B, C, 4A and 5), giving credence to the idea that this feature is required for medaka sperm compatibility. However, finger 2 N-glycosylation is not sufficient on its own nor in combination with the tested amino acid changes to enable medaka sperm compatibility with an otherwise zebrafish Bncr protein (Fig. 5F, G), indicating that additional features are also required.

To date, only three sperm proteins (Dcst1, Dcst2, and Spaca6) have been reported as essential for fertilization in zebrafish[9,11], yet none of them have been shown to act as Bncr's interaction partner. Although the identity of Bncr's interaction partner remains elusive, this study provides valuable insights into the amino acid sites and protein features within Bncr that are needed for binding sperm, thereby shedding light on what is required by the unknown interaction partner. To reconcile the observations that the medaka and zebrafish Bncr interaction partners exhibit asymmetrical specificity yet can interact with the fugu Bncr protein with comparable efficiency, we propose three possible explanations. Either the medaka and zebrafish interaction partners are entirely different molecules, or the binding sites for the two species' interaction partners on Bncr are different with unequal binding affinities. Alternatively, the zebrafish Bncr interaction partner on sperm may have an overall higher expression that results in higher avidity even when presented with a suboptimal Bncr with lower affinity. Such a strategy would ensure efficient sperm binding to the egg with risk of binding to heterospecific eggs, which is consistent with the ability of zebrafish and carp to hybridize with each other and more distantly related fish species[56,57,70,71]. Identification of Bncr's interaction partner(s) on sperm will enable differentiating between these possibilities to reveal the molecular nature of Bncr's essential lock-and-key mechanism.

## Methods

### Ethics statement

All animal experiments were conducted according to Austrian and European guidelines for animal research and approved by the Amt der Wiener Landesregierung, Magistratsabteilung 58—Wasserrecht (animal protocols GZ 342445/2016/12 and MA 58-221180-2021-16 for work with zebrafish; animal protocol GZ: 198603/2018/14 for work with medaka).

### Zebrafish and medaka husbandry

Zebrafish (*Danio rerio*) were raised according to standard protocols (28 °C water temperature; 14/10 h light/dark cycle). TLAB fish, generated by crossing zebrafish AB with stocks of the natural variant TL (Tüpfel long fin), served as wild-type zebrafish for all experiments. Wild-type medaka (*Oryzias latipes*, CAB strain) were raised according to standard protocols (28 °C water temperature; 14/10 h light/dark cycle) and served as wild-type medaka. *Oryzias curvinotus* were raised under the same conditions. *Bouncer* mutant zebrafish and medaka Bouncer-expressing transgenic zebrafish lines have been published previously[15].

### Generation of medaka *bncra* and *bncrb* mutants

Medaka *bncra* and *bncrb* mutants were generated using Cas9-mediated mutagenesis. Guide RNAs (sgRNAs) targeting the third (*bncra*) and second exons (*bncrb*) (Table 1) were synthesized by in vitro transcription using the MEGAscript T7 Transcription Kit (Thermo Fisher) after annealing oligos according to[72]. *Cas9* mRNA was synthesized using the mMESSAGE mMACHINE SP6 Transcription Kit (Thermo Fisher) using a linearized pCS2 vector template containing the Cas9 ORF[72]. One-cell medaka embryos (CAB strain) were co-injected with *cas9* mRNA and sgRNAs in 1X Yamamoto's ringer's solution (1.00 g NaCl, 0.03 g KCl, 0.04 g CaCl$_2$·2H$_2$O, 0.10 g MgCl$_2$·6H$_2$O, 0.20 g NaHCO$_3$ in 1000 ml, pH 7.3). Potential founder fish were crossed to wild-type CAB fish; the offspring from these crosses were screened by PCR for mutations in *bncra* (bncra_F: AGTACAAGCATCTGAGTAGGG and bncra_R: AGGCTGTGAACCTGACTG) and *bncrb* (bncrb_F: AGAG GCCTTTATAATGTGGACA and bncrb_R: CCATCTCATAGGAACCAC AGA) based on a shift in amplicon size compared to wild-type. Offspring of founder fish were raised to adulthood and in-crossed to produce homozygous mutants. The 5-nt and 38-nt deletions in exons 3 and 2, respectively, were detected by PCR and confirmed by Sanger sequencing to be frameshift mutations in *bncra* and *bncrb*, respectively. The wild-type and mutant sequences and corresponding translated amino acid sequences are provided in Supplementary Data File 1. Genotyping of *bncra* and *bncrb* mutants was done using PCR with the primers given above and standard gel electrophoresis using a 4% agarose gel.

### Quantification of medaka in vivo fertilization rates

To quantify fertilization rates of wild-type and mutant medaka, mating crosses were set up the night before inside the tanks in the fish water system. One male per two or three females was set up in the same tank; the male and females were separated with a vertical divider which was removed the morning of egg collection. After mating, the eggs were collected carefully with a fine mesh net using the thumb and forefinger to remove them from each female's body and placed into a separate petri dish containing 1X Yamamoto's ringer's solution. The eggs from each female were visually inspected under a dissection microscope and the number of unactivated eggs was recorded. These are easily distinguished from activated eggs based on their dark appearance, higher density of cortical alveoli, and the close apposition of the chorion to the egg membrane. Approximately 2–3 h post collection, the eggs were inspected again, and fertilization rates were quantified based on the presence of cell cleavage. Fertility rates for the same individuals were measured in at least biological triplicates.

### Generation of transgenic zebrafish lines

All zebrafish transgenic lines were generated using Tol2-mediated transgenesis. To generate plasmids encoding other fish Bncrs, chimeras, and ancestral states for transgenesis, each Bncr ORF lacking its endogenous signal peptide sequence but including the C-terminal tail was ordered as a custom gBlock (IDT) and Gibson cloned into a vector containing Tol2 sites, the *actb2* promoter, and the zebrafish Bncr signal peptide followed by sfGFP. Each Bncr sequence was inserted in frame downstream of sfGFP such that the resulting plasmids were as follows: Tol2—*actb2* promoter—zebrafish Bncr SP—sfGFP—Bncr sequence including C-terminal tail—SV40 UTR—Tol2. All amino acid substitution constructs were generated using PCR-based site-directed mutagenesis of plasmids containing the wild-type medaka or zebrafish Bncr ORF as the template. All transgene sequences are provided in Supplementary Data File 3. To generate transgenic zebrafish lines, *tol2* mRNA was co-injected with the plasmid encoding the desired Bncr transgene into one-cell stage zebrafish embryos from a ♀ *bncr*$^{+/-}$ x ♂ *bncr*$^{-/-}$ cross. Larvae were screened for fluorescence 1 day post fertilization (1 dpf) and grown to adulthood. Potential founders were crossed to *bncr*$^{+/-}$ or *bncr*$^{-/-}$ zebrafish and their progeny (F1) was grown to adulthood if fluorescent at 1 dpf. Homozygous *bncr* mutant F1 and F2 fish stably expressing the desired transgene were used for experimentation.

### Transgenic egg imaging with CellMask

Transgenic zebrafish females were set up with males as previously described. On the morning of collection, females were either allowed to mate with males or were squeezed according to the IVF protocol. Eggs were collected immediately in blue water (3 g Instant Ocean® sea salt per 10 l fish system water, 0.0001% (w/v) methylene blue) and allowed to activate for ~10 min. As soon as their chorions were lifted, 15–20 eggs were manually dechorionated with fine forceps in a Silguard dish filled with 1X Danieau's solution (58 mM NaCl, 0.7 mM KCl, 0.4 mM MgSO$_4$, 0.6 mM Ca(NO$_3$)$_2$, 5 mM HEPES, pH 7.6). Eggs were incubated at RT with gentle rocking for 15 min in a watch glass containing 0.01 mg/ml CellMask Deep Red plasma membrane stain (Invitrogen) in 250 µl 1X Danieau's solution and covered with aluminum foil. After

**Table 1 | Oligo sequences for sgRNAs and tracrRNA for generating medaka *bncra* and *bncrb* mutants**

| | |
|---|---|
| bncra_sgRNA1 | TAATACGACTCACTATAggTCTTCCATGCTGCTTTGCTGGTTTTAGAGCTAGAAATAGCAAG |
| bncra_sgRNA2 | TAATACGACTCACTATAggTTGCTACTACAGCCCCGTCCGTTTTAGAGCTAGAAATAGCAAG |
| bncrb_sgRNA1 | TAATACGACTCACTATAggAGGTGTTCCAGGGTAGAGACGTTTTAGAGCTAGAAATAGCAAG |
| bncrb_sgRNA2 | TAATACGACTCACTATAggCGACACTCGGTGGTGAAGTTGTTTTAGAGCTAGAAATAGCAAG |
| bncrb_sgRNA3 | TAATACGACTCACTATAgGCTCCTCGCCTCCATCCTGTGTTTTAGAGCTAGAAATAGCAAG |
| bncrb_sgRNA4 | TAATACGACTCACTATAggCCTCAGTCCTGTCTCTACCCGTTTTAGAGCTAGAAATAGCAAG |
| common tracr oligo | AAA AGC ACC GAC TCG GTG CCA CTT TTT CAA GTT GAT AAC GGA CTA GCC TTA TTT TAA CTT GCT ATT TCT AGC TCT AAA AC |

Guide RNA sequences targeting exon 3 (bncra) and exon 2 (*bncrb*) that were used to generate the corresponding mutants in medaka, as well as the common tracr oligo sequence used for synthesizing each complete sgRNA.

incubation, the eggs were transferred to a new watch glass containing 1X Danieau's solution and were then imaged immediately in an agarose mold filled with 1X Danieau's solution using an upright point laser scanning confocal microscope (LSM800 Examiner Z1, Zeiss) with a ×10/0.3 N-achroplan water objective.

## Quantification of *bncr* transgene copy number by qPCR

Primers targeting medaka *bncra* (medbncr_F: TCAGGTTCA-CAGCCTACGTC and medbncr_R: GTTACAGTACGGCCAGTCACA) and zebrafish *bncr* (zfbncr_F: CACCAGATGATCCGGGGAAA and zfbncr_R: CTGGGAGTTGCAGTAGTGTCC) were directly compared for efficiency by amplification of a dilution series of a template plasmid containing one copy of each transgene. The template plasmid was cloned for this purpose using a pBluescript II SK(+) vector (gift from Katharina Lust, Tanaka lab) and contained the following elements: I-SceI site−medaka *actb* promoter−zebrafish Bncr SP−mCherry−mature zebrafish Bncr including C-terminal tail−SV40 UTR−mature medaka Bncra including C-terminal tail−I-SceI site. Linear regression analysis of the standard curve yielded a line of best fit; the equation of which could be used to calculate copy number based on Cq value for each primer pair. Eggs from three medaka or zebrafish females per line were collected immediately after laying and homogenized in TRIzol (Thermo Fisher Scientific). Total RNA from each egg sample was isolated by standard phenol/chloroform extraction. cDNA synthesis was done using the iScript cDNA Synthesis Kit (Bio-Rad); for each sample, 500 ng of input RNA was used. qPCR was performed in technical duplicates using 2X GoTaq Master Mix (Promega) and the primers given above depending on the target transgene. Copy number was then calculated based on the average Cq value of technical duplicates for each biological replicate using the equation derived for each primer pair.

## In vitro fertilization assay with medaka and zebrafish

To collect zebrafish eggs and sperm for in vitro fertilization (IVF), wild-type TLAB zebrafish males were set up the night before experimentation with transgenic or wild-type zebrafish females in a small, plastic breeding tank with a divider separating the two fish. On the day of experimentation, sperm was collected from zebrafish males after anesthetization in 0.1% (w/v) tricaine (25X stock solution in dH₂O, buffered to pH 7−7.5 with 1 M Tris pH 9.0) in fish system water. Using plastic tubing with a capillary in one end and a pipette filter tip in the other end, sperm was mouth-pipetted from the urogenital opening of each male placed belly-up in a slit in a sponge wetted with fish water. Sperm was transferred directly to a 1.5-ml tube containing Hank's saline (0.137 M NaCl, 5.4 mM KCl, 0.25 mM Na₂HPO₄, 0.44 mM KH₂PO₄, 1.3 mM CaCl₂, 1.0 mM MgSO₄, 4.2 mM NaHCO₃) on ice. To collect zebrafish eggs, transgenic or wild-type zebrafish females were anesthetized on the day of experimentation as described above. After anesthetization, the abdomen of each female was carefully dried on a paper towel and the fish was transferred to a petri dish. Gentle pressure was applied to the belly of the female with the thumb of one hand while her back was supported with a finger of the other hand. Approximately 50−100 eggs were exuded from the female before transferring her to a second petri dish and repeating the process. The fish was immediately placed back into fish water, and one clutch of eggs was fertilized with zebrafish sperm, and the other with medaka sperm, such that the same number of sperm were used on both clutches of eggs. In total, 500 μl of blue water was added immediately to each clutch after sperm addition to activate gametes. The dishes were left undisturbed for 3−5 min and then filled with blue water and placed into an incubator at 28 °C. Functionality of each batch of both zebrafish and medaka sperm used in these experiments was confirmed by conducting control IVF with wild-type conspecific eggs for both species. IVF with medaka sperm and eggs was performed as described in ref. 32.

In general, based on the number of egg clutches of eggs to be fertilized in each experiment, one male was used per 100 μl of Hank's

saline. Because sperm is used in great excess during IVF, any concentration above 50,000 sperm/μl was used. Sperm were counted manually in a Neubauer chamber to ensure that the same number of medaka and zebrafish sperm was used on each sample in the same experiment. In general, 3−4 million sperm were used to fertilize each clutch of eggs. Fertilization rates were quantified ~3 h after IVF by using a dissection microscope and counting the number of fertilized embryos with cell cleavage and unfertilized eggs that had remained at the one-cell stage and did not develop. For all transgenic lines, at least two (in most cases at least five) individuals were tested for fertility in vivo and in IVF in biological triplicates unless a certain individual died between trials or failed to give eggs. In such cases, additional individuals were tested.

## Positive selection analysis

Mature Bncr protein sequences were aligned using MAFFT[73] and codon alignment was generated using PAL2NAL[74]. Codon alignments were then used as input into IQ-TREE[75] to generate the best substitution model and a maximum-likelihood tree was generated using 1000 ultrafast bootstrap iterations[76]. Codon alignments and the maximum-likelihood tree were used as input into HyPhy[77] to test the mode of selection acting on Bncr in fish. A suite of tests was performed across all sequences by using MEME[78], FUBAR[79], FEL[80], and BUSTED[81]. The mode of selection acting on the zebrafish and medaka lineages was tested by selecting on the branches leading to these lineages and performing aBSREL[82] and Contrast-FEL[83]. We further mapped the residues identified to be under selection regimes of interest onto the predicted 3-D structures of zebrafish and medaka Bncr[40,41]. In addition, the level of conservation was mapped onto the 3-D structures for zebrafish and medaka Bncr using CONSURF[84] and visualized using PyMOL (http://www.pymol.org).

## Prediction of Bncr ancestral states

Bncr amino acid sequences were aligned using MUSCLE[85] with default parameters. A phylogeny was then reconstructed using MrBayes and mcmc = 2,000,000[86]. Finally, ancestral amino acid states were reconstructed for all nodes of the obtained phylogeny with PAML with default parameters[87]. The alignment, phylogeny, and ancestral reconstructions as well as the relevant control files are available on GitHub (https://github.com/kristabriedis/AncestralBncrs)[88].

## De-glycosylation and western blot analysis

To collect egg cap lysates for de-glycosylation enzyme treatment and western blot analysis, transgenic zebrafish females for each line of interest and a male were set up the night before in a mating tank and separated with a plastic divider. On the morning of collection, the fish were allowed to mate, and their eggs were collected immediately in blue water. As soon as the chorions were lifted, the eggs were de-chorionated and de-yolked manually with fine forceps in a Silguard dish filled with 1X Danieau's solution. Eggs were dissected 5 at a time such that a total of 20 egg caps were collected in 8 μl of 1X Danieau's solution and were immediately pipetted into a tube on dry ice. Samples were kept at −70 °C until processing. To each sample, 32 μl of nuclease-free water was added and samples were divided equally into two tubes. The untreated sample was kept at −70 °C, while the other was treated overnight with Protein Deglycosylation Mix II (NEB) using non-denaturing reaction conditions according to the manufacturer's protocol. All samples were boiled with 1X Laemmli buffer containing 10% ß-mercaptoethanol before SDS-PAGE using Mini-PROTEAN TGX (Bio-Rad) pre-cast gels. After SDS-PAGE, samples were wet-transferred onto a nitrocellulose membrane which was blocked with 5% milk powder in 0.1% Tween-20 in 1X TBS (TBST). Membranes were incubated in primary rabbit anti-GFP antibody [1:1000, (A11122, Invitrogen)] overnight at 4 °C, then washed with TBST before HRP-conjugated goat anti-rabbit secondary antibody [1:10,000 (115−036−045, Dianova)]

incubation for 30 min to 1 h. Membranes were washed a few times in TBST before HRP activity was visualized using Clarity Western ECL Substrate (Bio-Rad) on a ChemiDoc (Bio-Rad). For visualizing tubulin levels, membranes were stripped using Restore Western Blot Stripping Buffer (Thermo Fisher Scientific) before washing, blocking, and incubation with mouse anti-alpha-tubulin antibody [1:20,000 (T6074, Merck)] and proceeding with HRP-conjugated goat anti-mouse secondary antibody staining [1:10,000 (115–036–062, Dianova)] and detection as described above. Uncropped images of Western blots are provided in the Source Data file.

## Statistical analysis

Statistical comparisons between medaka *bncra* and *bncrb* mutants vs. wild-type and the transgenic *bncra* rescue line were performed using the Kruskal–Wallis test with Dunn's multiple comparisons test at the 0.95 confidence level. Statistical comparisons between clutches of eggs fertilized by medaka vs. zebrafish sperm in IVF experiments were made using the Wilcoxon matched-pairs signed rank test (two-tailed) with the method of Pratt, such that pairs for which the two values were equal were not excluded. The median of the bias indices derived from paired IVF data was tested for being significantly different from a hypothetical median of 0 (indicating no bias) using the Wilcoxon signed rank test (two-tailed) at the 0.95 confidence level. The method of Pratt was used such that median values equal to 0 were not excluded.

## Reporting summary

Further information on research design is available in the Nature Portfolio Reporting Summary linked to this article.

## Data availability

All data needed to evaluate the conclusions in the paper are presented in the paper and/or the Supplementary Information. Accessions for all sequences referred to within the paper are provided in Supplementary Data Files. All biological materials generated for this study can be obtained from the corresponding author without any restrictions. Source data are provided with this paper.

## Code availability

Files pertaining to ancestral state reconstruction are available on GitHub (https://github.com/kristabriedis/AncestralBncrs)[88].

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

## Acknowledgements

We thank Manfred Schartl for sharing RNA-seq data from medaka ovaries and testes prior to publication; Maria Novatchkova for help with RNA-seq analysis; Katharina Lust for advice on medaka techniques; Milan Malinsky for input on Lake Malawi cichlid Bouncer sequences; Felicia Spitzer, Mirjam Binner, and Anna Bandura for help with genotyping; Friedrich Puhl, Kerstin Rattner, Julia Koenig, and Dijana Sunjic for taking care of zebrafish and medaka; and the Pauli lab for helpful discussions about the project and feedback on the manuscript. K.R.B.G. was supported by a DOC Fellowship from the Austrian Academy of Sciences. Work in the Pauli lab was supported by the FWF START program (Y 1031-B28 to A.P.), the ERC CoG 101044495/GaMe, the HFSP Career Development Award (CDA00066/2015 to A.P.), a HFSP Young Investigator Award (RGY0079/2020 to A.P.) and the FWF SFB RNA-Deco (project number F80). The IMP receives institutional funding from Boehringer Ingelheim and the Austrian Research Promotion Agency (Headquarter grant FFG-852936). Work by J.S. and Y.M. in this project was supported by the Israel Science Foundation grant 636/21 to Y.M. Work by L.J. was supported by the Swedish Research Council grant 2020-04936 and the Knut and Alice Wallenberg Foundation grant 2018.0042. For the purpose of Open Access, the author has applied a CC BY public copyright license to any Author Accepted Manuscript (AAM) version arising from this submission.

## Author contributions

K.R.B.G. and A.P. conceived the study; K.R.B.G. designed, performed, and analyzed experiments, with assistance from K.P. in some IVF experiments, in vivo fertilization rate collection, and western blotting, and B.S.S. in some IVF experiments. A.S. conducted phylogenetic analysis. J.S. and Y.M. performed positive selection analysis. L.J. provided structural predictions and insights into glycosylation patterns of Bouncer. F.K. generated predicted ancestral states of Bouncer. A.P. supervised the study. K.R.B.G. wrote the original draft. K.R.B.G. and A.P. revised the manuscript with input from all authors.

## Competing interests

The authors declare no competing interests.
