## [Peer review file · Nature Communications]

REVIEWER COMMENTS

Reviewer #1 (Remarks to the Author):

During the past several years, gamete interactions and gamete fusion in organisms across multiple taxa have been shown to depend on members of broadly conserved protein families. Although exciting, these advances raise a conundrum: How can a conserved protein family be at the heart of a species-specific event? These workers have exploited their ability to work with fertilization in multiple species of fish to approach an answer. Using rigorous *in vitro* and *in vivo* assays for fertilization in combination with structure-guided mutagenesis strategies, these authors identify amino acid residues in 3 conserved loops in the fish oocyte membrane protein Bouncer that are heart of the species-specificity of gamete interactions in organisms across multiple taxa.

Earlier studies in invertebrates also addressed species-specificity of fertilization-related proteins, but they focused on proteins that function at steps before membrane interactions. And, it was not possible to genetically manipulate the systems to test models. These current studies focus directly on gamete membrane proteins and the findings are of greater depth and much finer resolution than any previous studies. They represent a substantial amount of painstaking work spanning molecules and fish tanks, the experiments are carefully and rigorously executed, and the results are compelling.

I believe, however, that the authors are missing an opportunity to place their work in the larger context of fertilization across the domains of life. Indeed, the first two sentences of the abstract and the introduction give the impression that the authors are addressing questions about fertilization *per se*, and not restricting their consideration to certain taxa. In keeping with this broad perspective, the introduction also brings in fertilization-related findings from vertebrates (fish and mammals) and invertebrates (echinoderms and mollusks). Given that the manuscript addresses these big questions on fertilization and species-specificity of fertilization-essential protein families, the introduction and discussion should also bring in findings on other broadly conserved, fertilization-related proteins.

One such fertilization-related protein is the primordial gamete fusogen, HAP2. Although not yet identified in chordates or fungi, HAP2 is essential for gamete fusion in unicellular organisms and green plants as well as in many multicellular animals, including arthropods and mollusks. The Johnson lab (Wong et al., 2010; doi:10.1371/journal.pgen.1000882) examined the fusion competence of *Arabidopsis thaliana* HAP2 chimeras containing regions of HAP2 from rice and other plants. Similarly, while not examining function, studies of the X-ray structures of HAP2 from *Arabidopsis*, the unicellular green alga, *Chlamydomonas reinhardtii*, and the pathogenic protist *Trypanosoma cruzi*, by the Johnson and Rey labs (Fedry et al., 2019; doi.org/10.1371/journal.pbio.2006357) reported on the evolutionary diversification of HAP2 membrane insertion motifs. Additionally, members of the FUS1/GEX2 family are essential for the gamete membrane adhesion that precedes fusion in *Chlamydomonas* and in *Arabidopsis*, and its

conservation across plant lineages indicates that it is yet another conserved protein family that is likely to participate in species-specific interactions. Finally, the KAR5/GEX1/BMB family of nuclear envelope proteins is essential for fusion of gamete pronuclei in the zygote in organisms across the domains of life, including in the malaria organism *Plasmodium berghei* and the organism of focus here, *Danio rerio*.

By providing this larger context, that evolution has used conserved domain architectures for species-specific functions, readers will have a much greater appreciation of the importance and elegance of the finding in this manuscript.

Additional, more minor comments are below.

Abstract: I found it difficult to understand the term "gamete compatibility." It seems to be imprecise, non-mechanistic, and limiting. As indicated above, this manuscript addresses evolutionary changes in proteins that determine species-specificity and provides a basis for mechanistic answers.

Line 29: Meaning of phrase "species-specific between medaka and zebrafish" is difficult to understand?

Line 84: The phrase "general species-specific factor" is confusing? If a factor is specific, it doesn't seem that it could also be general? Do the authors mean that they investigated whether Bncr was sufficient to control species-specificity of gamete interactions? Sentence needs to be re-written to more effectively communicate the idea being proposed.

Line 111 +: To avoid confusion in this section and others where different species and their wt and mutant gametes are being discussed, it would be helpful to always indicate the species. I recognize that it should be clear without the names, but readers will not be as intimately familiar with names and designations as the authors.

Line 146+: Many readers of Nature Communications might not be familiar with the evolutionary relationships of the organisms being considered. Thus, authors should present a brief description of evolutionary distance between the multiple species whose Bncrs are used here. Such a description will allow the reader to better appreciate the biological significance of these findings.

Line 163: As the text later in the paragraph alludes to, these experiments do not establish bona fide asymmetry. It seems that to establish such asymmetry would require that zebrafish sperm be tested with medaka fish whose eggs were expressing zebrafish bncr.

Line 222: Should be "These data."

Line 265: For readers unfamiliar with these approaches, need to describe the evidence supporting the statement that Bncr's evolution is dominated by pervasive purifying (negative) selection.

Reviewer #2 (Remarks to the Author):

In the first half of the paper, the authors confirmed that deletion of medaka Bouncer (Bncra, not Bcrb), an essential factor for gamete recognition in bony fishes, causes female infertility and that there is interspecific incompatibility through the Bncr. In the second half, focusing on species-specific compatibility between medaka and zebrafish fertilization, the authors derived the answer that N-glycan in finger 2 of Bncr for medaka and 45th and 63rd arginine residues in finger 2 and 3 of Bncr, respectively, for zebrafish are important for cross-species compatibility based on evolutionary and genetic mutagenesis approaches.

Overall, the study is clearly presented and the results are significant for this research field, but unfortunately, the question will not be resolved until its counterpart(s) (sperm Bouncer receptor) is found. As the authors discussed, they may be employing entirely different binding partners. If so, there must be a different regulatory system between medaka and zebrafish fertilization in terms of gamete compatibility. Therefore, I do not think that this manuscript meets the criteria for publication at this stage.

Major questions:

Is it possible to break the barrier of interspecies incompatibility by performing in vitro and in vivo fertilization using zebrafish oocytes carrying the R45A/R63L mutation plus N-glycosylation of only finger 2 of Bncr and medaka spermatozoa?

Is it possible that the recombinant protein of the Bncr ectodomain binds directly to spermatozoa? If so, it should be evaluated using its various mutants to prove whether gamete compatibility is governed by a single molecule on the ovum side.

Reviewer #3 (Remarks to the Author):

The manuscript "Divergent molecular signatures in Bouncer define cross-fertilization boundaries" by Gert et al. provides a thorough structure/function analysis of the Bouncer protein to determine what domains, amino acids, and post-translational modifications are important for the species-specific function of this protein when compared between zebrafish and medaka. This work contains two important contributions to the field. First, by expanding the number of species examined, the authors show that the species specificity previously observed between zebrafish Bouncer and medaka Bouncer does not exist amongst all fish species tested, highlighting the importance of studying fertilization across a multitude of organisms and showing that the molecular features that make the Bouncer protein zebrafish compatible or medaka compatible are not mutually exclusive. Second, this type of detailed molecular analysis of fertilization proteins is critical to our ability to build models and hypotheses about the mechanism of action for these proteins. Overall, the study is well done, the data are laid out in a clear manner, and the conclusions are supported by the data.

I have 2 minor suggestions for the authors.

1) For the *bnkra* and *bncrb* transgenic rescues, could the authors clarify the details of the constructs regarding whether the *Bncr* ORF was the cDNA sequence (no introns) or some other isoform specific ORF sequence?

2) The authors mention several times asymmetry in the specificity of zebrafish sperm interaction with Bouncer and medaka sperm interaction with Bouncer. While they offer good explanations for these observations, they tend to focus on one-on-one interactions between Bouncer and an (currently unknown) interacting protein on the sperm. It would be helpful to add some brief discussion that fertilization is more likely to occur in the context of multiple protein interactions between the gametes rather than one-to-one ligand-receptor interactions. I think it's important to note that while they are swapping out Bouncer, the rest of the egg proteins are zebrafish egg proteins and this could influence the overall success of the zebrafish sperm and include this in the overall interpretation of their data. Likewise, Bouncer may interact with other proteins on the egg surface and/or multiple interacting partners on the sperm surface. The authors do a nice job focusing on the molecular details of Bouncer but it would be good to then place these results in the larger context of multiple protein interactions mediating fertilization and how they may or may not contribute to or influence the species specificity of the process.

Reviewer #4 (Remarks to the Author):

This paper describes a gamete incompatibility that may in part explain reproductive isolation between different species of fish. This incompatibility is based on variation in the egg membrane protein Bouncer, which allows fertilization to occur across species boundaries in some species crosses but not others. This mechanism is of clear relevance to the evolution and maintenance of reproductive isolation between fish species, and as such, has likely played a major role in shaping the evolution of fish biodiversity.

My primary area of expertise is evolutionary genetics - I study speciation, hybridization, and reproductive isolation in fish. My comments are therefore in that context, and restricted to the evolutionary context of this paper.

Major comments:

Overall, I think the evolutionary aspects of this paper are underdeveloped. There is (appropriately) a lot of focus on details of the molecular pathways that underpin the function of Bouncer, but the evolutionary framing is not as prominent as I think it should be to explain the significance of these interesting findings about evolutionary variation in Bouncer. I suggest incorporating more of the speciation and hybridization literature. A major question in recent years has been why hybridization occurs readily in some crosses but not others; this paper presents one very compelling answer, but the context is not completely described.

A broad evolutionary pattern that has been observed is that likelihood of hybridization is inversely proportional to genetic distance (e.g. Bolnick & Near 2005, *Evolution*). Fig. 4B suggests pretty broad phylogenetic compatibility of Bouncer, which is somewhat at odds with this observation. Some discussion of how fertilization compatibility through Bouncer is likely to interact with other aspects of reproductive isolation would strengthen both the introduction and the discussion.

RESPONSE TO REVIEWERS' COMMENTS

Reviewer #1 (Remarks to the Author):

During the past several years, gamete interactions and gamete fusion in organisms across multiple taxa have been shown to depend on members of broadly conserved protein families. Although exciting, these advances raise a conundrum: How can a conserved protein family be at the heart of a species-specific event? These workers have exploited their ability to work with fertilization in multiple species of fish to approach an answer. Using rigorous *in vitro* and *in vivo* assays for fertilization in combination with structure-guided mutagenesis strategies, these authors identify amino acid residues in 3 conserved loops in the fish oocyte membrane protein Bouncer that are heart of the species-specificity of gamete interactions in organisms across multiple taxa.

Earlier studies in invertebrates also addressed species-specificity of fertilization-related proteins, but they focused on proteins that function at steps before membrane interactions. And, it was not possible to genetically manipulate the systems to test models. These current studies focus directly on gamete membrane proteins and the findings are of greater depth and much finer resolution than any previous studies. They represent a substantial amount of painstaking work spanning molecules and fish tanks, the experiments are carefully and rigorously executed, and the results are compelling.

We kindly thank the reviewer for this positive assessment of our work.

I believe, however, that the authors are missing an opportunity to place their work in the larger context of fertilization across the domains of life. Indeed, the first two sentences of the abstract and the introduction give the impression that the authors are addressing questions about fertilization *per se*, and not restricting their consideration to certain taxa. In keeping with this broad perspective, the introduction also brings in fertilization-related findings from vertebrates (fish and mammals) and invertebrates (echinoderms and mollusks). Given that the manuscript addresses these big questions on fertilization and species-specificity of fertilization-essential protein families, the introduction and discussion should also bring in findings on other broadly conserved, fertilization-related proteins.

One such fertilization-related protein is the primordial gamete fusogen, HAP2. Although not yet identified in chordates or fungi, HAP2 is essential for gamete fusion in unicellular organisms and green plants as well as in many multicellular animals, including arthropods and mollusks. The Johnson lab (Wong et al., 2010; doi:10.1371/journal.pgen.1000882) examined the fusion competence of *Arabidopsis thaliana* HAP2 chimeras containing regions of HAP2 from rice and other plants. Similarly, while not examining function, studies of the X-ray structures of HAP2 from *Arabidopsis*, the unicellular green alga, *Chlamydomonas reinhardtii*, and the pathogenic protist *Trypanosoma cruzi*, by the Johnson and Rey labs (Fedry et al., 2019; doi.org/10.1371/journal.pbio.2006357) reported on the evolutionary diversification of HAP2 membrane insertion motifs. Additionally, members of the FUS1/GEX2 family are essential for the gamete membrane adhesion that precedes fusion in *Chlamydomonas* and in *Arabidopsis*, and its conservation across plant lineages indicates that it is yet another conserved protein family that is likely to participate in species-specific interactions. Finally, the KAR5/GEX1/BMB family of nuclear envelope proteins is essential for fusion of gamete pronuclei in the zygote in organisms across the domains of life, including in the malaria organism *Plasmodium berghei* and the organism of focus here, *Danio rerio*.

By providing this larger context, that evolution has used conserved domain architectures for species-specific functions, readers will have a much greater appreciation of the importance and elegance of the finding in this manuscript.

We are grateful to the reviewer for providing this useful feedback with which to better frame our findings in the broader context of fertilization across eukaryotes. Following this reviewer's

recommendation, we have added HAP2/GCS1 to the introduction and have expanded our discussion on both HAP2 and other conserved fertilization-related proteins. In the discussion, we have furthermore elaborated on conserved protein domains that have recurrently been observed in fertilization-related proteins across taxa. In light of Bncr's being a three-finger protein, we decided to discuss the three-finger domain of Ly6/uPAR proteins more thoroughly. As we have observed with Bncr, this domain is an adaptable protein module that enables species-specific gamete interactions in some combinations yet is also broadly compatible across phylogeny. Bringing in Bncr's mammalian homolog, SPACA4, we further comment on how Bncr/SPACA4 has diverged both in terms of cell type expression and function in fertilization from fish to mammals, and how this further demonstrates the flexibility of the three-finger domain to be adapted to specific reproductive contexts, potentially similar to the evolutionary diversification of HAP2's fusion loops in plants and trypanosomes. By expounding on the importance of these conserved domains in fertilization and placing Bncr into this context, we trust that the impact of our results will be more readily recognized.

Additional, more minor comments are below.

Abstract: I found it difficult to understand the term "gamete compatibility." It seems to be imprecise, non-mechanistic, and limiting. As indicated above, this manuscript addresses evolutionary changes in proteins that determine species-specificity and provides a basis for mechanistic answers.

We appreciate the reviewer's comment to explain better what we mean by "gamete compatibility." We have clarified this term by rephrasing it as, "molecular compatibility between gametes," and further explain in the abstract that this refers to the idea that a sperm and egg can recognize and bind each other via their surface proteins.

Line 29: Meaning of phrase "species-specific between medaka and zebrafish" is difficult to understand?

We thank the reviewer for raising this point and helping to increase the clarity of our paper. We have changed this phrase to say, "The egg membrane protein Bouncer confers species specificity to gamete interactions between medaka and zebrafish, preventing their cross-fertilization."

Line 84: The phrase "general species-specific factor" is confusing? If a factor is specific, it doesn't seem that it could also be general? Do the authors mean that they investigated whether Bncr was sufficient to control species-specificity of gamete interactions? Sentence needs to be re-written to more effectively communicate the idea being proposed.

We thank the reviewer for pointing this out and providing helpful feedback on our phrasing. We have rewritten this sentence as follows: "...we investigated whether other fish species' Bncr proteins also mediate species-specific gamete interaction..." to clarify that we wanted to investigate whether Bncr binds sperm species-specifically across fish species as we had made the previous observation of Bncr's species specificity between only zebrafish and medaka.

Line 111 +: To avoid confusion in this section and others where different species and their wt and mutant gametes are being discussed, it would be helpful to always indicate the species. I recognize that it should be clear without the names, but readers will not be as intimately familiar with names and designations as the authors.

We thank the reviewer for making us aware that our current notation could be confusing for readers. We have improved these sections by indicating the species as suggested.

Line 146+: Many readers of Nature Communications might not be familiar with the

evolutionary relationships of the organisms being considered. Thus, authors should present a brief description of evolutionary distance between the multiple species whose Bncrs are used here. Such a description will allow the reader to better appreciate the biological significance of these findings.

We are grateful to the reviewer for this useful suggestion. We have added a supplementary figure panel (2A) illustrating the evolutionary distances between the species whose Bncr proteins are being studied to make this more readily understood by the reader.

Line 163: As the text later in the paragraph alludes to, these experiments do not establish bona fide asymmetry. It seems that to establish such asymmetry would require that zebrafish sperm be tested with medaka fish whose eggs were expressing zebrafish bncr.

We thank the reviewer for bringing up this important point and have included more detail in the manuscript to explain observations from a separate but related study. In our other study, currently on bioRxiv (<https://www.biorxiv.org/content/10.1101/2021.11.03.467109v1>), we show that wild-type medaka eggs expressing zebrafish Bncr can indeed be fertilized by zebrafish sperm as long as activation is artificially induced by addition of a calcium ionophore, calcimycin (see the figure below from the *bioRxiv* manuscript, left).

We have added more detail to the current manuscript to explain this observation, and stress that in the other study and as mentioned in the current manuscript, zebrafish sperm were observed to fertilize even wild-type medaka eggs without zebrafish Bncr expression (albeit at a low rate of 2% on average). This has never been observed in the opposite orientation as medaka sperm cannot fertilize wild-type zebrafish eggs nor those overexpressing zebrafish Bncr. These observations are in line with the idea that zebrafish sperm appear to have a laxer threshold for successful Bncr interaction compared to medaka sperm such that zebrafish sperm can fertilize both medaka and zebrafish eggs expressing medaka Bncr at endogenous or overexpression levels, but this is not true for medaka sperm and zebrafish Bncr.

We have now additionally performed these experiments in *bncr*^{-/-} medaka eggs expressing zebrafish Bncr and show that these eggs can neither be activated nor fertilized by medaka sperm, whereas zebrafish sperm can fertilize them when they are activated artificially (figure above, right). Thus, whatever advantage medaka sperm would have by being presented with a conspecific egg is overruled by the absence of medaka Bncr and cannot be substituted for by zebrafish Bncr. This is in contrast to our observations in zebrafish, in which zebrafish *bncr*

eggs expressing medaka Bncr can be fertilized by zebrafish sperm, though at a low rate (see Fig. 2A, <20% *in vitro*). Because these experiments are directly relevant for the generation of zebrafish-medaka hybrids which is the focus of our other study, we would like to include these data in the manuscript focusing on hybrids rather than this one.

Because it is technically challenging to control all variables when performing cross-species IVF with both zebrafish and medaka sperm and eggs, we have limited our statements in the paper to a suggestion of asymmetry rather than a definite conclusion. Furthermore, even by comparing Bouncer-sperm interaction on zebrafish eggs vs. medaka eggs, intrinsic differences in micropylar morphology and size (in addition to other protein-protein interactions) might also influence the ability of sperm to reach the membrane to the same efficiency. The medaka micropyle is narrower and longer than that of zebrafish, thus it is more likely that medaka and zebrafish sperm are on an equal playing field when traversing the wider, shorter zebrafish micropyle, whereas medaka sperm may be at an advantage for navigating the narrower medaka micropyle. On top of this, medaka and zebrafish eggs have separate mechanisms for induction of egg activation. Medaka eggs require sperm contact to undergo activation, most likely due to the action of phospholipase C zeta 1 (PLCZ1) provided by the sperm (Uwa, 1987; Iwamatsu, 1989; Iwamatsu, 1992). In contrast, zebrafish eggs are activated upon contact with water even in the absence of sperm. Zebrafish sperm are unable to activate medaka eggs, likely due to the lack of a *plcz1* homolog in the zebrafish genome. Thus, performing IVF with zebrafish sperm and medaka eggs requires the addition of a calcium ionophore (calcimycin) to trigger egg activation post sperm addition. Because it is difficult to mimic the exact physiological timing of egg activation in conjunction with sperm binding in this way, fertilization rates with zebrafish sperm are likely lower than they would be due to inefficient or poorly timed egg activation for some eggs.

Our data presented later in the paper further supports the idea of asymmetry as zebrafish sperm exhibit compatibility with most Bncr proteins tested, while medaka sperm are compatible only with Bncr proteins that have finger 2 N-glycosylation (see Fig. 4A and Fig. 5). These observations provide a possible explanation for this asymmetry: medaka sperm have a strict requirement for a specific Bncr N-glycosylation pattern, while zebrafish sperm maintain compatibility with multiple Bncr variants bearing many different N-glycosylation patterns. An important comparison to highlight is that while zebrafish sperm are compatible with wild-type zebrafish Bncr (glycosylated on fingers 1 and 3) as well as a non-glycosylated zebrafish Bncr variant (Herberg et al., 2018), medaka sperm cannot function with medaka Bncr as soon as its normal glycosylation on finger 2 is changed, whether it be removed completely or moved to fingers 1 and/or 3 (Fig. 5). These comparisons were all made on the zebrafish egg membrane, giving some intrinsic benefit to zebrafish sperm interaction, however, this advantage can be overridden by medaka-compatible Bncr expression and it is clear that the repertoire of Bncr variants functional with zebrafish sperm is much larger than is functional with medaka sperm, owing at least in part to the specific medaka requirement for finger 2 N-glycosylation that is not shared by zebrafish sperm.

Line 222: Should be "These data."

Thank you for pointing out this mistake; we have changed the text accordingly.

Line 265: For readers unfamiliar with these approaches, need to describe the evidence supporting the statement that Bncr's evolution is dominated by pervasive purifying (negative) selection.

We appreciate this valuable feedback and thank the reviewer for pointing out another aspect that will enhance the readability of our manuscript. We have included more details regarding the evidence that demonstrates that Bncr's evolution is dominated by negative selection in the

corresponding paragraph and also added a general explanation of positive and negative selection.

Reviewer #2 (Remarks to the Author):

In the first half of the paper, the authors confirmed that deletion of medaka Bouncer (Bncra, not Bncrb), an essential factor for gamete recognition in bony fishes, causes female infertility and that there is interspecific incompatibility through the Bncr. In the second half, focusing on species-specific compatibility between medaka and zebrafish fertilization, the authors derived the answer that N-glycan in finger 2 of Bncr for medaka and 45th and 63rd arginine residues in finger 2 and 3 of Bncr, respectively, for zebrafish are important for cross-species compatibility based on evolutionary and genetic mutagenesis approaches. Overall, the study is clearly presented and the results are significant for this research field, but unfortunately, the question will not be resolved until its counterpart(s) (sperm Bouncer receptor) is found. As the authors discussed, they may be employing entirely different binding partners. If so, there must be a different regulatory system between medaka and zebrafish fertilization in terms of gamete compatibility. Therefore, I do not think that this manuscript meets the criteria for publication at this stage.

We thank the reviewer for recognizing the impact of our work and for raising this important point regarding Bouncer's interaction partner. We agree that identifying the interaction partner(s) on sperm is necessary for ultimately understanding how Bouncer and its interactor(s) mediate sperm-egg binding in a species-specific manner. However, identifying protein-protein interactions between the sperm and egg membranes is challenging for multiple reasons: (1) these interactions are between membrane proteins which are difficult to study biochemically due to their transmembrane domains; (2) interactions between cell-surface proteins are often weak and transient, and thus difficult to detect; and (3) the proteins themselves are not necessarily highly expressed. This is illustrated by the fact that in vertebrates, the only sperm-egg interaction pair identified thus far is IZUMO1 and JUNO. Discovery of JUNO as the egg receptor for IZUMO1 nearly 10 years after the identification of IZUMO1 was only achieved by oligomerizing the ectodomain of IZUMO1 to increase its avidity as a binding probe (Bianchi et al., 2014). While new factors essential for fertilization such as SPACA6, DCST1/2, and TMEM95 have all been identified in recent years (Noda et al., 2020; Barbaux et al., 2020; Lamas-Toranzo et al., 2020; Inoue et al., 2021; Noda et al., 2022; Binner et al., 2022), their interactors and mechanisms of action remain open questions.

Nevertheless, the fertilization field has been moved forward tremendously by important studies that have characterized and provided new insights into multiple aspects of previously identified fertilization factors, including their evolution, dynamics, and protein structures (Moi et al., 2022, *Nat. Commun.*; Brukman et al., 2022, *J. Cell Biol.*; Grayson, 2015, *R. Soc. Open Sci.*; Vance et al., 2022, *Commun. Biol.*; Tang et al., 2022, *PNAS*; Matsumura et al., 2022, *Front. Cell Dev Biol.*; Inoue and Wada, 2022, *Biol. Reprod.*; Nakajima et al., 2022, *Sci. Rep.*). We therefore hold that the identity of Bouncer's sperm interaction partner is outside the scope of this manuscript. Our manuscript provides both important evolutionary insights and structure-function analyses of Bouncer, ultimately identifying key amino acids and post-translational modifications that provide specificity for zebrafish vs. medaka sperm interaction. Our study will furthermore greatly expedite characterization of the binding interface for Bouncer and its sperm interaction partner(s) once found.

Major questions:

Is it possible to break the barrier of interspecies incompatibility by performing in vitro and in vivo fertilization using zebrafish oocytes carrying the R45A/R63L mutation plus N-glycosylation of only finger 2 of Bncr and medaka spermatozoa?

We thank the reviewer for this insightful suggestion. We have generated additional zebrafish *bncr*^{-/-} lines expressing transgenic zebrafish Bncr rescue constructs that have only finger 2 N-glycosylation combined with either R63L alone or both R63L and R45A. However, these

changes enabled only a very poor rescue with medaka and zebrafish sperm *in vitro* (Figure 5G) though compatibility with zebrafish sperm was still possible *in vivo* and expression was confirmed by imaging sfGFP signal on the egg membrane (Figure 5H, Suppl. Figure 5H). We furthermore tested whether providing zebrafish Bncr with both finger 2 N-glycosylation and the entire sequence of medaka finger 3 was sufficient to enable medaka sperm compatibility, but the results were similar to the two aforementioned lines (Figure 5G-H), indicating that additional features must be required for breaking the interspecies incompatibility barrier. We have included this additional data in the revised manuscript.

Is it possible that the recombinant protein of the Bncr ectodomain binds directly to spermatozoa? If so, it should be evaluated using its various mutants to prove whether gamete compatibility is governed by a single molecule on the ovum side.

We thank the reviewer for this valuable suggestion. We had initially planned to test Bncr's compatibility either via transgenic lines *in vivo* (this work is presented here) or as an alternative strategy via recombinant protein expression. While our strategy using transgenic lines expressing different variants of Bncr proteins has been highly successful and is presented in this manuscript, we have faced technical challenges in obtaining even the wild-type version of recombinant zebrafish and medaka Bncr proteins over the past several years, making it infeasible to perform such experiments, which would require the production of multiple different recombinant protein variants. Our *in vivo* experiments have shown that replacing only zebrafish Bncr with medaka Bncr on the zebrafish egg is indeed sufficient on its own to enable medaka sperm compatibility, effectively demonstrating that even if a co-factor or additional egg proteins were needed for sperm binding, Bncr alone is decisive for compatibility. Moreover, this conclusion is also directly confirmed by data obtained for a related manuscript on *bioRxiv* (Gert et al., 2021; <https://www.biorxiv.org/content/10.1101/2021.11.03.467109v1>), in which we demonstrate that expressing zebrafish Bncr on medaka eggs enables zebrafish sperm to fertilize them (see figure below), again confirming that only Bncr is required on the side of the egg to mediate gamete compatibility between zebrafish and medaka.

Reviewer #3 (Remarks to the Author):

The manuscript "Divergent molecular signatures in Bouncer define cross-fertilization boundaries" by Gert et al. provides a thorough structure/function analysis of the Bouncer protein to determine what domains, amino acids, and post-translational modifications are important for the species-specific function of this protein when compared between zebrafish and medaka. This work contains two important contributions to the field. First, by expanding the number of species examined, the authors show that the species specificity previously observed between zebrafish Bouncer and medaka Bouncer does not exist amongst all fish species tested, highlighting the importance of studying fertilization across a multitude of organisms and showing that the molecular features that make the Bouncer protein zebrafish compatible or medaka compatible are not mutually exclusive. Second, this type of detailed molecular analysis of fertilization proteins is critical to our ability to build models and hypotheses about the mechanism of action for these proteins. Overall, the study is well done, the data are laid out in a clear manner, and the conclusions are supported by the data.

We thank the reviewer for this positive assessment of our work and its impact on the field.

I have 2 minor suggestions for the authors.

1) For the *bncra* and *bncrb* transgenic rescues, could the authors clarify the details of the constructs regarding whether the Bncr ORF was the cDNA sequence (no introns) or some other isoform specific ORF sequence?

Thank you for pointing out this important detail and helping to improve our manuscript. We have updated the text to specify that these rescue constructs are encoded by the cDNA sequences.

2) The authors mention several times asymmetry in the specificity of zebrafish sperm interaction with Bouncer and medaka sperm interaction with Bouncer. While they offer good explanations for these observations, they tend to focus on one-on-one interactions between Bouncer and an (currently unknown) interacting protein on the sperm. It would be helpful to add some brief discussion that fertilization is more likely to occur in the context of multiple protein interactions between the gametes rather than one-to-one ligand-receptor interactions. I think it's important to note that while they are swapping out Bouncer, the rest of the egg proteins are zebrafish egg proteins and this could influence the overall success of the zebrafish sperm and include this in the overall interpretation of their data. Likewise, Bouncer may interact with other proteins on the egg surface and/or multiple interacting partners on the sperm surface. The authors do a nice job focusing on the molecular details of Bouncer but it would be good to then place these results in the larger context of multiple protein interactions mediating fertilization and how they may or may not contribute to or influence the species specificity of the process.

We thank the reviewer for this insightful feedback that will help contextualize our results in the molecular framework of fertilization as a whole. In the current version of the manuscript, we have mentioned in the results that there may be intrinsic bias for zebrafish sperm given the fact that all transgenes tested were expressed on zebrafish eggs. We have now elaborated further and state that there is a possible influence of other, currently uncharacterized sperm-egg interactors that would be species-matched for zebrafish sperm but not for medaka. We have furthermore expounded on this idea in the discussion and acknowledge the potential effects of suboptimal interactions between other, unidentified sperm and egg proteins on zebrafish eggs and medaka sperm. However, we stress that changing only zebrafish Bncr to a medaka-compatible Bncr is sufficient on its own to enable medaka sperm to fertilize zebrafish eggs, highlighting the fact that Bncr is the decisive factor for compatibility in this species combination.

This conclusion is also directly confirmed by data obtained for our related manuscript on *bioRxiv* (Gert et al., 2021; <https://www.biorxiv.org/content/10.1101/2021.11.03.467109v1>), in which we demonstrate that expressing zebrafish Bncr on wild-type medaka eggs enables zebrafish sperm to fertilize them as long as they are artificially activated by addition of a calcium ionophore, calcimycin (see the figure below from the *bioRxiv* manuscript, left). We have now performed additional experiments for the other manuscript demonstrating that *bncr*^{-/-} medaka eggs expressing zebrafish Bncr can be fertilized by zebrafish sperm, but not by medaka sperm (figure below, right). These data support the idea that asymmetry exists between medaka and zebrafish Bncr-sperm interaction as no background fertilization was observed with medaka sperm and *bncr*^{-/-} medaka eggs expressing zebrafish Bncr (figure below, right), while zebrafish sperm can fertilize *bncr*^{-/-} zebrafish eggs expressing medaka Bncr, though at a low rate (see Fig. 2A, <20% *in vitro*). This further confirms that only Bncr is required on the side of the egg to mediate gamete compatibility between zebrafish and medaka.

Reviewer #4 (Remarks to the Author):

This paper describes a gamete incompatibility that may in part explain reproductive isolation between different species of fish. This incompatibility is based on variation in the egg membrane protein Bouncer, which allows fertilization to occur across species boundaries in some species crosses but not others. This mechanism is of clear relevance to the evolution and maintenance of reproductive isolation between fish species, and as such, has likely played a major role in shaping the evolution of fish biodiversity.

We thank the reviewer for this positive assessment of our work and its impact on the field.

My primary area of expertise is evolutionary genetics - I study speciation, hybridization, and reproductive isolation in fish. My comments are therefore in that context, and restricted to the evolutionary context of this paper.

Major comments:

Overall, I think the evolutionary aspects of this paper are underdeveloped. There is (appropriately) a lot of focus on details of the molecular pathways that underpin the function of Bouncer, but the evolutionary framing is not as prominent as I think it should be to explain the significance of these interesting findings about evolutionary variation in Bouncer. I suggest incorporating more of the speciation and hybridization literature. A major question in recent years has been why hybridization occurs readily in some crosses but not others; this paper presents one very compelling answer, but the context is not completely described.

A broad evolutionary pattern that has been observed is that likelihood of hybridization is inversely proportional to genetic distance (e.g. Bolnick & Near 2005, Evolution). Fig. 4B suggests pretty broad phylogenetic compatibility of Bouncer, which is somewhat at odds with this observation. Some discussion of how fertilization compatibility through Bouncer is likely to interact with other aspects of reproductive isolation would strengthen both the introduction and the discussion.

We thank the reviewer for providing this useful feedback regarding our manuscript and its evolutionary context. We have stated the implications our results may have for hybridization in fish in the introduction and have expanded our discussion to include insights on how Bncr's cross-species compatibility may be important for hybridization and biodiversity of fish in general. We cite specific examples of hybridization between both closely and distantly related species and how high sequence identity between their Bncr orthologs likely permits this on the molecular level. We furthermore include an example of reproductive isolation via sexual selection in haplochromine cichlids and how Bncr compatibility allows hybridization between these species upon changing environmental conditions, thereby illustrating the influence of cross-species Bncr compatibility in the context of other reproductive isolation mechanisms. We speculate that Bncr-mediated cross-species compatibility may have impacted speciation in certain fish lineages in such ways.

REVIEWERS' COMMENTS

Reviewer #1 (Remarks to the Author):

The authors have appropriately addressed my concerns. The revised discussion, with more information about and consideration of the evolutionary aspects of these findings, substantially enriches the manuscript. Now the reader can better appreciate that the in-depth molecular “tinkering” (DOI: 10.1126/science.860134) by these authors to create hybrids in the laboratory provides an unprecedented look into the tinkering during evolution that generated new species.

(NB: In line 333 need to change “this data supports” to “these data support.”)

Reviewer #2 (Remarks to the Author):

The issue of Bouncer receptor on the sperm side remains, however I think that the author has adequately remedied all concerns with the modifications.

I am satisfied with the revisions provided by the author.

Reviewer #3 (Remarks to the Author):

I am satisfied with the authors' revisions and responses to review. I have no further comments.